# Mixture of Nested Experts: Adaptive Processing of Visual Tokens

**Gagan Jain**[◇*]  **Nidhi Hegde**[◇]  **Aditya Kusupati**[◇†]
**Arsha Nagrani**[◇]  **Shyamal Buch**[◇]  **Prateek Jain**[◇]  **Anurag Arnab**[◇]  **Sujoy Paul**[◇*]

[◇]Google DeepMind  [†]University of Washington

{jaingagan,sujoyp}@google.com

## Abstract

The visual medium (images and videos) naturally contains a large amount of information redundancy, thereby providing a great opportunity for leveraging efficiency in processing. While Vision Transformer (ViT) based models scale effectively to large data regimes, they fail to capitalize on this inherent redundancy, leading to higher computational costs. Mixture of Experts (MoE) networks demonstrate scalability while maintaining same inference-time costs, but they come with a larger parameter footprint. We present Mixture of Nested Experts (MoNE), which utilizes a nested structure for experts, wherein individual experts fall on an increasing compute-accuracy curve. Given a compute budget, MoNE learns to dynamically choose tokens in a priority order, and thus redundant tokens are processed through cheaper nested experts. Using this framework, we achieve equivalent performance as the baseline models, while reducing inference time compute by over *two-fold*. We validate our approach on standard image and video datasets - ImageNet-21K, Kinetics400, and Something-Something-v2. We further highlight MoNE's adaptability by showcasing its ability to maintain strong performance across different inference-time compute budgets on videos, using only a single trained model.

## 1 Introduction

Visual tokens, the fundamental building blocks of image and video representations, often exhibit strong inter-dependencies, spatially in images and spatio-temporally in videos. This offers a potential avenue for optimization in visual processing, as processing every token with equal emphasis may not be necessary for achieving optimal results. Traditional Vision Transformer (ViT) [20] and Video Vision Transformer (ViViT) [2] based models, however, process all tokens with equal emphasis, disregarding this inherent codependency and leading to unnecessary computational burden. This becomes a major bottleneck when deploying these models in real-world scenarios, where computational resources may be limited and real-time processing is required.

To this end, conditional computation has become a promising line of research to increase the capacity of a network, while only conditionally activating a part of it during inference. Sparse Mixture of Experts (MoEs) was initially popularized for Natural Language Processing (NLP) [42, 22],but it has been gaining attention for furthering conditional computation ideas in vision [39, 1, 35, 50] as well. While MoEs bring in improved performance at a given inference cost, they also increase the overall parameter count, leading to increased storage requirements. Moreover, these works rely on experts that have the same parameter count and compute, limiting their ability to reduce computational costs without resorting to skipping tokens entirely.

---

[*]equal contribution

38th Conference on Neural Information Processing Systems (NeurIPS 2024).

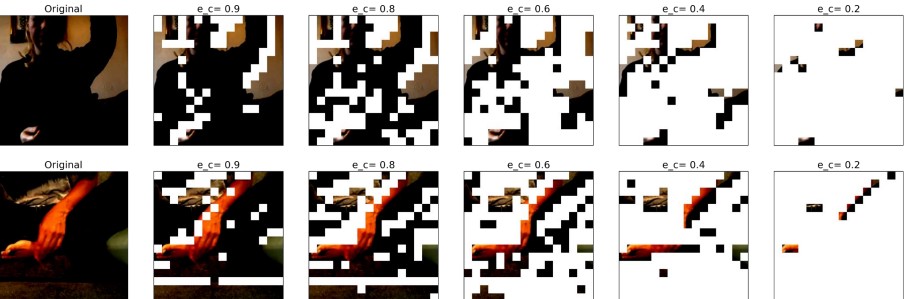

Figure 1: **MoNE's learned token importance:** From *left to right*, fewer image tokens are processed using the full model – to fit a compute budget – by an increasing threshold on MoNE's router logits.

In this work, we devise the **Mixture of Nested Experts (MoNE)** framework, which provides a scalable approach to conditional computation, bringing in significant reductions at inference time, while working with the same parameter space as the baseline model. MoNE draws inspiration from nested architectures [47, 31, 53], particularly MatFormer [19], that learns multiple representations of the same data with varying levels of details, based on structured slices of the parameter space. MoNE employs these structured nested models as experts in the MoE framework (without increasing parameter count), and learns a network to route tokens to these experts. We explore various design choices and present an effective recipe for allocating compute to experts, assigning tokens to experts, and training the MoNE framework. For the assignment operation, we propose Expert Preferred Routing (EPR), a routing algorithm that greedily assigns tokens to experts under capacity constraints based on router predictions. Figure 1 shows token importance as perceived by MoNE. We propose the following **three primary contributions**:

1. We introduce the novel Mixture of Nested Experts (MoNE) framework to dynamically allocate computational resources for Vision Transformer (ViT) based models.

2. Given a fixed parameter count, MoNE offers the flexibility of learning networks at much lower FLOPs (~ 2.3× on video datasets) and real-time latency (~ 2×), while being quality neutral.

3. Rigorous experiments show that MoNE works well for both image and video transformers, and visualizations depict that tokens routed to larger experts correlate well with regions of interest.

## 2  Related Work

Transformers [45] have become the de-facto architecture for processing data across multiple modalities spanning language [10, 36], images [20, 17], video [2, 49] and audio [23] and combinations thereof [38]. Consequently, there have been numerous efforts to improve the efficiency of transformers to make them more amenable for deployment in real-world applications [44]. These include approaches like efficient approximations of attention [12, 48], local attention [32, 4, 13] and reducing the number of tokens in the transformer [40, 29, 8] among others. Our work focuses on conditional computation [5, 34], observing that some input tokens are easier to process than others, and therefore require less computation during inference.

Mixtures-of-Experts (MoE) transformers learn to route tokens to one of multiple expert MLPs [42, 22]. Although such models conditionally process input tokens, each expert has the same parameter- and FLOP-count, meaning that the total computation is constant for each input. More relevant to our approach, Mixture of Depths [37] extends the routing logic of MoE to conditionally skip an expert completely, thus total computation for each input varies dynamically. Completely skipping tokens being a hard unretrievable decision, our work chooses from an array of nested network, which effectively process information and help to stabilize training by getting rid of discontinuities.

Nested architectures [47, 31, 53] on the other hand, learn hierarchical representations of the input, where the first $k$ hidden dimensions encode the most relevant information. This allows to extract multiple models with varying inference compute from a single trained model, similar to 'Mix-n-Match' in [19]. However, these models do not process tokens adaptively. Our model, in contrast, consists of a learned router which dynamically routes tokens to experts of different hidden dimensions based on the given compute constraints. Therefore, instead of requiring the user to select the hidden dimensions of

each transformer layer, our model only needs a single compute constraint input. Moreover, we show experimentally the superior accuracy-efficiency trade-offs achieved by our approach.

We note that other conditional computation approaches include "early exiting" [46, 41, 21, 28] such that the processing of "easy inputs" terminates before passing through all layers of the transformer. In addition, the ACT [25] algorithm was proposed for recurrent neural networks, and uses a "ponder cost" to learn a "halting score" for when to stop processing a particular input. This has since been extended to recurrent transformers [15], and also to each individual token in a transformer [52, 51], thus adaptively determining which tokens in a transformer to process. In contrast, our approach does not drop tokens, rather processes them with smaller nested models. This allows us to retain most of the information, and hence dampen the effect of irrecoverable decisions. We experimentally verify that our adaptive approach offers strong compute-performance trade-offs. Flextron [11] is a concurrent work, which looks at elastic inference, specified by user latency needs, with a focus on language modeling. Unlike Flextron, MoNE is guaranteed to learn models bounded by the specified latency needs and is able to learn from a single training phase, without using a surrogate model.

## 3  Preliminaries

Here, we discuss the concept of *nested models*, on which we build Mixture of Nested Experts (MoNE), followed by a discussion about Mixture of Nested Experts (MoE), and its differences from MoNE.

### 3.1  Nested Models

For the purposes of this work, we use the Vision Transformer (ViT) [20] as an example of a full model, from which nested submodels can be derived. Inspired by MatFormer [19], we define these submodels for every layer of the network, for both Self-Attention and MLP (see Appendix A.1). The key idea is that in a feature projection operation $\mathbf{W}\mathbf{x}$, where $\mathbf{W} = \left[\mathbf{W}_{[:\frac{D}{m}]}, \mathbf{W}_{[\frac{D}{m}:]}\right]$, and $\mathbf{W}_{[:\frac{D}{m}]}$ denotes "slicing" the first $\frac{D}{m}$ dimensions, we can extract a partial projection $\mathbf{W}_{[:\frac{D}{m}]}\mathbf{x}_{[:\frac{D}{m}]}$. This can be done for any projection in the transformer, and we can extract smaller models from it. We refer to these as nested models, and $D/m$ as the nested model dimension. This is shown in Figure 2a. The *Extract* operation extracts the first $D/m$ features and applies the corresponding projection sub-matrix to it, while the *Pad* operation pads it back to full dimension $D$ before residual connections and LayerNorm. While MatFormer applies the nested structure only to the hidden dimension of the MLP layer, in our approach we extend it to the in- and out-projections of both the Self-Attention (SA) and MLP layer. In the SA block, irrespective of the sub-model used in the in-projections, it is always projected to the model dimension $D$ for the $(\mathrm{QK}^T)\mathrm{V}$ operation. The same thing is performed in MLP, where the hidden dimension is always $4D$, as in ViT, irrespective of in/out-projection dimension.

We extract $E$ nested models with exponentially-spaced model dimensions. Therefore, for a typical value of $E = 4$, the model dimension for the nested models are $\left[\frac{D}{8}, \frac{D}{4}, \frac{D}{2}, D\right]$. Note that while we build upon the idea of nested models from MatFormer, we do not share their training strategy which involves joint optimization through a weighted loss over these submodels. In contrast, we treat these nested models as distinct experts with varying compute requirements. The Mixture of Nested Experts (MoNE) framework (described in detail in Sec. 4.1) then dynamically routes input tokens to these nested experts based on their information content, with the idea that more informative tokens should be processed by larger (and thus more computationally expensive) nested models.

### 3.2  Mixture of Experts

A Mixture of Experts (MoE) layer in a transformer can be represented as $\mathrm{MoE}(\mathbf{x}) = \sum_{i=1}^{E} g(\mathbf{x})_i e_i(\mathbf{x})$, where $E$ is the number of experts, $e_i()$ are the expert models each having their own parameters, $g : \mathbb{R}^D \to \mathbb{R}^E$ is the routing/gating function, which decides the experts which should process $\mathbf{x}$. Note that $g$ is sparse with only $k << E$ non-zero terms. During inference, only those experts are active.

MoE strictly increases the parameter count, but maintains the same inference FLOPs by setting $k = 1$. However, it still needs to process all tokens with the same pre-defined compute. In contrast, in MoNE, we do not extend the parameter count of the model, due to the nesting structure (see Sec. 3.1), and dynamically choose a nested expert during inference. Unlike in MoE, where all experts have the same capacity, in MoNE with $k = 1$ always, $e_i \subset e_{i+1}$, which allows us to dynamically allocate compute.

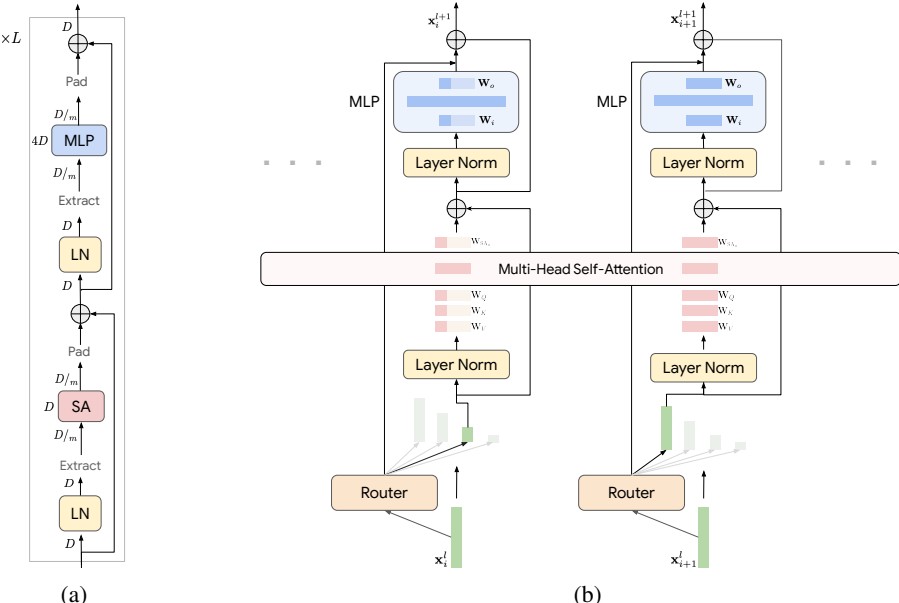

Figure 2: (a) **Nested model:** Partial in- and out-projections in the SA and MLP layers create nested models. $m$ controls the parameter count and the FLOPs of nested models. The self-attention information exchange happens at the full model dimension $D$, MLP dimension is set to $4D$ as in ViT. (b) **Mixture of Nested Experts (MoNE)**: Each token $\mathbf{x}$ is routed to a nested network, denoted by different model dimension in the diagram. Here $\mathbf{x}_i$ gets routed to a nested model with model dimension $D/4$, whereas $\mathbf{x}_{i+1}$ gets to the full model. The information exchange between these tokens of different dimension happens in the self-attention block, where they are always projected to the same dimension. The router weights are also multiplied with the features for proper flow of gradients. A lighter color in the weight matrix indicate a sliced matrix to construct the nestedness.

## 4 Methodology

In this section, we describe the details of our Mixture of Nested Experts (MoNE) framework for efficient inference. We assume a Vision Transformer (ViT) [20] based architecture for our approach, and then extend it to Video ViT (ViViT) [2] as well.

### 4.1 Mixture of Nested Experts (MoNE)

**Tokenization:** In this paper, as our primary focus is images and videos, the model input is in $\mathbb{R}^{H \times W \times 3 \times T}$, where $T = 1$ for images and $T > 1$ for videos. After tokenization, the input to the transformer is $\mathbf{X} \in \mathbb{R}^{D \times N}$ where $N$ is the number of tokens, and $D$ their model dimension. For images, we have $N = H/p_h \cdot W/p_w$, and for video, $N = T/p_t \cdot H/p_h \cdot W/p_w$, where $H, W, T$ are the input height, width and duration respectively. $p_h$, $p_w$ and $p_t$ are the patch sizes along these respective dimensions. We use the ViT [20] and ViViT [2] architectures to tokenize images and videos respectively, obtaining a list of tokens $\mathbf{X} = \{\mathbf{x}_i\}_{i=1}^{N}$.

**MoNE Block:** The Mixture of Nested Experts (MoNE) framework is a dynamic routing mechanism that processes visual tokens using nested models with varying computational capacities, instead of processing all tokens with the full model. A pictorial repsentation of the model is presented in Figure 2b. Let $\mathcal{B}^l = \{\mathcal{B}_1^l, \ldots, \mathcal{B}_E^l\}$ denote the nested blocks at a certain layer $l$ with increasing parameter sizes, $\mathcal{B}_E^l(.)$ being the full model block. A router network decides the appropriate nested block to use for every token. Hence information from tokens of different model dimension interact with each other. This is enabled by performing self-attention at the full model dimension $D$ as discussed before. For each token $\mathbf{x}_i$, a router produces a probability distribution over the $E$ nested experts, $\mathbf{r}_i = \text{softmax}(\mathbf{W_r}\mathbf{x}_i + \mathbf{b_r})$, where $\mathbf{W_r}$ and $\mathbf{b_r}$ denote the router weights and bias respectively.

These router predictions are sent to an assignment algorithm, which assigns every token to a single appropriate nested expert. Based on the assignments, we update the features for the $i^{th}$ token in the $l^{th}$ layer as follows -

$$\mathbf{x}_i^{l+1} = \mathbf{z}_i^l + \left(\alpha \mathbf{r}_{i,j}^l + 1\right) \cdot \mathcal{B}_j^{\text{FFN},l}(\mathbf{z}_i^l) \qquad \mathbf{z}_i^l = \mathbf{x}_i^l + \mathcal{B}_j^{\text{SA},l}(\mathbf{x}_i^l) \qquad (1)$$

where the $j^{th}$ nested expert is chosen by the Expert Preferred Router [EPR(.)] algorithm for the $i^{th}$ token as per Eq. 2:

$$j^* = \text{EPR}\big(i; \{\mathbf{r}_i^l\}_{i=1}^N\big) \tag{2}$$

Note that the multiplication of the router predictions with the model output in Eq. 1 allows gradient propagation through the router weights. We also introduce a learnable parameter $\alpha \in [0, 1)$, initialized to 0, which ensures proper gradient flow during the initial training stages, specifically during finetuning from a pre-trained MatFormer model. Without scaling, a low initial router prediction would dampen the block output, whereas the initial multiplicative factor being 1 ensures a stable starting point.

**Features and Loss:** The feature of the last layer $\mathbf{x}_i^L$ is used for downstream applications. For classification tasks, we apply global average pooling on all the token features and apply a linear classifier layer to predict the categories.

## 4.2  Token to Nested Expert Assignments

Within the MoNE framework, the routing strategy is crucial for achieving an optimal balance between performance and computational efficiency. Traditionally there are two primary routing strategies – token choice [42] and expert choice [39] . In token-choice routing, the router predicts the probability distribution over the available experts, and picks the expert with the highest probability. However, this can suffer from load balancing issues, with most of the tokens being routed to one or few experts. Hence, inference time compute is only bounded by the compute of the full model. On the other hand, in expert choice routing, each expert selects the top-$k$ tokens with the highest preference for that expert. This guarantees perfect bounds on computation. Potential conflicts due to token selection by multiple experts are resolved by prioritizing based on model size.

Formally, we consider a given distribution of nested models applied to the tokens, represented as $\mathbf{c} = \{c_1, \ldots, c_E\}$, s.t., $\sum_i c_i = 1$, which we call the capacity distribution over the nested models. The method for obtaining a suitable capacity distribution, given the inference time compute requirements, will be discussed in Sec. 4.3. Given router probabilities $\mathbf{r}_i$ for $N$ tokens across $E$ experts, we employ an Expert Preferred Routing algorithm (Algorithm 1). This is a greedy assignment approach that gives higher preference to larger nested models, aiming to identify the most important tokens first. We begin by examining the router predictions for the biggest to the smallest model, assigning $k_j = \lfloor c_j N \rfloor$ of the remaining tokens to $j^{th}$ nested model. Any remaining tokens, arising from integer packing constraints, are assigned to the smallest model. Algorithm 1 presents the proposed Expert Preferred Routing (EPR) algorithm.

---

**Algorithm 1** Expert Preferred Routing (EPR)

---

**Require:** $\mathbf{r} \in \mathbb{R}^{E \times N}$ (router predictions), $\mathbf{c}$ (capacity distribution, s.t., $\mathbf{c}^T \mathbf{1} = 1$),
**Ensure:** $M \in \{1, \ldots, E\}^N$ (nested model index)
1:   $M \leftarrow \mathbf{1}_N$                              Default assignments to the smallest model
2: **for** $j = E$ to $1$ **do**
3:      $k_j \leftarrow \lfloor c_j \cdot N \rfloor$
4:      $I \leftarrow \text{Top-}k\text{-Index}(\mathbf{r}[j, \ldots], k_i)$               Returns value and indices of Top-K
5:      $M[I] \leftarrow j$
6:      $\mathbf{r}[:, I] \leftarrow 0$                               Null out assigned ones
7: **end for**
8: **return** $M$

---

## 4.3  Capacity Distribution Across Experts

The Expert Preferred Routing (EPR) as described in Section 4.2 needs the individual expert's capacity bounds $c_i$ to be specified. To get this, we define a metric called the effective capacity : $e_c = \sum_{i=1}^E c_i d_i / D$, where $d_i = D/2^{E-i}$ is the model dimension of the $i^{th}$ nested model. Given a certain inference FLOP requirement, we can translate that to an equivalent effective capacity $e_c$. Since every token gets processed through exactly one nested expert, this along with the given budget imposes two constraints on the unknown capacity distribution $\mathbf{c}$. However, since the individual expert capacities vary log-linearly, multiple distributions $\mathbf{c}$ can lead to the same $e_c$ for $E > 2$ and it is non-trivial to choose

one over the other. MoEs generally use auxilliary loss functions [39, 42] to promote equal usage of experts. But in MoNE, that would render a certain fixed capacity, missing out on the flexibility that the framework provides to function with any capacity (as depicted later in Figure 5b). Hence, we invoke intuitive constraints to solve for **c**. Specifically, we incentivize the usage of larger models, while also adding an entropy term to ensure uniformity of capacities across experts. Given these constraints, we solve the following optimization problem:

$$
\begin{aligned}
\text{maximize} \quad & \sum_{i=1}^{E} \frac{c_i}{\delta^{i-1}} - \beta \sum_{i=1}^{E} c_i \cdot \log c_i \\
\text{subject to} \quad & \sum_{i=1}^{E} c_i = 1 \qquad \sum_{i=1}^{E} \frac{c_i}{2^{E-i}} = e_c \qquad 0 \le c_i \le 1 \quad \forall i \in \{1, ..., E\} \\
\text{given} \quad & 0 < e_c < 1, \quad E, \delta > 1, \quad \beta > 0
\end{aligned}
\tag{3}
$$

In practice, we set $(\beta, \delta)$ to $(10, 2)$ and use a Sequential Least SQuares Programming (SLSQP) optimizer to solve Eq. 3 for the capacity distribution **c**, which is then used by EPR (Algorithm 1) to get token to expert mappings. We empirically verify these choices in Section 6.

### 4.4 Videos

MoNE can be seamlessly adapted for video-based tasks. In videos, there exists another dimension – time – which adds to the significant redundancy in the tokens. Given the large number of tokens that can be obtained from a video, the computational costs grow drastically. To tackle this problem, works in literature factorize computation along space and time [2, 6], perform local windowed computation [33], etc. MoNE being a token based approach, directly extends to video encoders.

For video processing, we leverage the Factorized Encoder architecture of ViViT [2]. This architecture employs two distinct transformers: spatial and temporal. After tokenization, each temporal index yields a set of tokens representing information from local spatio-temporal neighborhoods. These spatial tokens interact within their temporal index for $L_s$ layers, culminating in a single global token per index. Subsequently, a temporal transformer processes these global tokens across $L_t$ layers. Given that the spatial transformer significantly dominates computational costs in this model, we integrate MoNE into the spatial component while maintaining full capacity for the temporal transformer. The router predicts expert assignments for all temporal frames independently, which are then consumed by the EPR(.) algorithm to produce frame-wise expert assignments.

## 5 Results

In this section, we empirically evaluate MoNE on multiple datasets spanning images and videos for different model sizes, assess its adaptability to stringent FLOP constraints, and depict real-time latency gains achieved by MoNE during inference.

**Implementation details:** We empirically evaluate MoNE on image and video classification. For image classification, we train the network with random initialization. As for video classification, we follow previous literature and start from a pre-trained MatViT [19] model due to the inherent nested structure required in MoNE. We follow the joint training strategy of MatViT, with separate losses an all model granularities. We implement MoNE on JAX [9] using BigVision [7] for image classification and Scenic [16] for video classification. We follow the AugReg [43] training strategy to train all our image classification models. For video classification tasks, we inherit all augmentations and hyperparameter values directly from the ViViT [2] paper.

For all experiments in this section, we place a single router at the first transformer layer, and propagate the router decisions to all the layers. We also multiply the router predictions (Eqn 1) to all layers, which ensures differentiable paths through the router network in all layers and allows the more evolved features from later layers to influence router learning. We also perform analysis of router placement in Section 6.

**Baselines:** We first compare with MatViT's nested models. As mentioned in the paper [19], we perform joint training over all four nested models that we consider in this work - $\{\frac{D}{8}, \frac{D}{4}, \frac{D}{2}, D\}$. MatViT is equivalent to MoNE, with a deterministic router to pass all tokens to the same nested

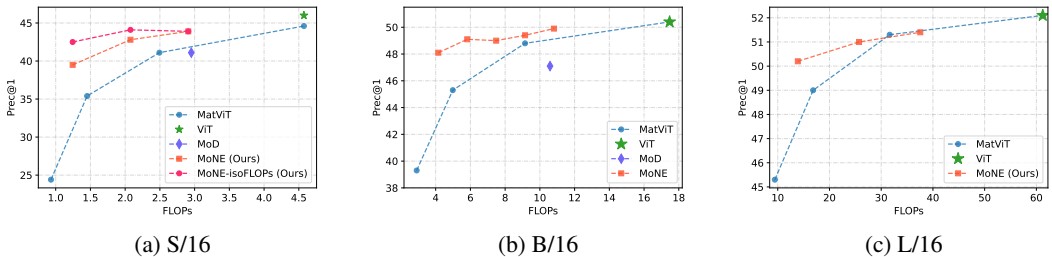

| (a) S/16 | (b) B/16 | (c) L/16 |

Figure 3: **Image classification:** Performance comparison of MoNE with baselines on ImageNet-21k for different model sizes. MoNE performs significantly better than MatViT and Mixture-of-Depth (MoD) and even benefits from isoFLOPs training (see fig a).

model. We show that adaptively mixing tokens with different model dimensions performs much better across datasets and tasks. We also compare with Mixture of Depths (MoD) [37], which is also a token routing algorithm, but proposed for language tasks. MoD takes the extreme decision of either processing or skipping for every token in a layer. MoNE, on the other hand, makes fuzzy decisions to choose intermediate-sized models, instead of skipping, which helps to retain significant information at the expense of low compute. We adopt the best reported MoD configuration: processing 12.5% of tokens every other layer while processing all tokens in the remaining layers.

We also emphasize that MoNE acts as a complementary framework to traditional MoEs like Sparse VMoE [39], and inference-time optimization techniques like Token Merging (ToMe) [8]. We present an extended discussion and further results in Appendix A.2, comparing with other adaptive baselines and validating the compounded savings by applying ToMe on MoNE.

**Images:** First, we evaluate MoNE on ImageNet-21k [18] classification using ViT. We experiment with S, B, and L models to showcase the efficacy of MoNE across model sizes. As ImageNet-21k can have multiple labels for an image, we report the commonly used precision@1 metric. Figure 3 shows the results for all the models on ImageNet-21k. MoNE performs much better than MatViT's nested models and MoD, specifically in the low FLOPs regimes. MoNE achieves comparable performance to baselines with around 2× reduction in FLOPs.

Following the literature on language models [37, 27], we experimented with isoFLOPs training, which involves training for the same number of FLOPs as the baseline models. Since MoNE models have fewer FLOPs compared to their ViT counterparts, they require more training epochs to achieve the same total training FLOPs. We conducted this experiment on the S/16 model (see Figure 3a) and observed additional improvements in MoNE's performance, particularly for the lower FLOPs models.

**Videos:** Since video models rely on heavy pre-training [2], we first train a baseline model with nested structure on the benchmark datasets - Kinetics-400 [31] and Something-Something-v2 (SSv2) [24]. We use the ViViT Factorized Encoder B/16 model [2] for our experiments and consistently report the 8x1 test accuracy, averaging predictions over 8 temporal clips [2]. Figure 4 illustrates the results of the MoNE framework, significantly outperforming the individual nested models. MoNE offers 2 − 3× reduction in FLOPs compared to the ViViT baseline, without any accuracy drop (On SSv2, the

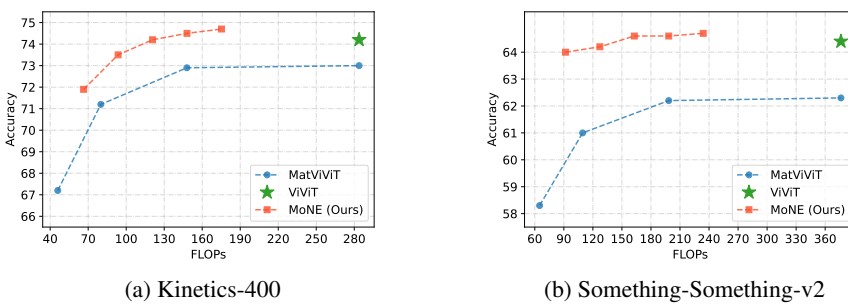

| (a) Kinetics-400 | (b) Something-Something-v2 |

Figure 4: **Video classification**: MoNE vs. baselines on video datasets. Finetuning with the isoFLOPs training regime leads to matching baseline with > 2× FLOP improvement.

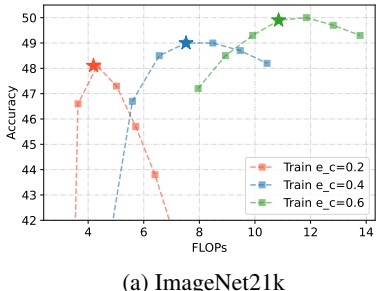

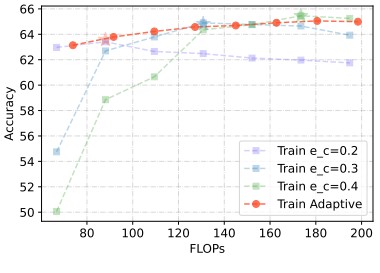

| (a) ImageNet21k | (b) Something-Something-v2 |

Figure 5: **Capacity adaptation during inference:** Performance changes when a model trained at a certain capacity (denoted as ★) is evaluated at other capacities. The "Train Adaptive" plot for SSv2 denotes a single model evaluated at different inference-time budgets.

FLOPs for MoNE are 162.8 vs 376.3, with similar accuracy – 64.6 vs 64.4). We always do isoFLOPs training while fine-tuning these models. We attribute the higher compute gains compared to images due to the greater (spatial and temporal) redundancy in videos, which MoNE exploits well.

**Inference time capacity adaptation:** Capacity adaptation during inference is crucial, as the inference time budget is often dynamic, changing based on user needs. Ideally, a model should adjust with little to no retraining. To evaluate this ability, we test how MoNE, trained at a specific effective capacity ($e_c$) performs when evaluated at other capacities. Fig. 5 presents the results for image and video classification. We observe that the model adapts well to nearby capacities. However, as expected, its ability declines with extreme shifts in the capacity budget between train and eval. The performance degradation is steeper while adapting a model trained at high capacity to low capacity. We also note that the performance degrades more gracefully in videos than on images, presumably due to the larger temporal redundancy.

To enhance model adaptability, we train a model with the capacity sampled uniformly at random from $\{0.15, 0.25, \ldots, 0.95\}$ at each training step. The results on SS-v2 (Figure 5b) demonstrate our framework's strong capability to adapt to any inference-time budget using a single model. It is interesting to note that the training FLOPs of this adaptively trained model are equal to those of a baseline model (isoFLOPs training). The model adapts extremely well even to capacities that are significantly different ($\{0.2, 0.3, \ldots\}$) from those sampled during training.

**Real Time Latency Gains**: In addition to the theoretical FLOP gains, Table 1 presents the real-time latency/throughput gains of MoNE-based ViViT model as compared to its baseline variant. The absolute wall clock times and throughput are compared on a single V100 GPU, achieving nearly two-fold improvement in both FLOPs as well as runtime, whilst maintaining accuracy.

Table 1: Real Time Latency and Throughput gains for MoNE on a single V100 GPU

| Method | FLOPs (G) | Throughput (clips/sec) | Latency (ms) | Top-1 Accuracy |
|---|---|---|---|---|
| ViViT-FE-B/16 | 376 | 15.8 | 129.2 | 64.4 |
| MoNE ($e_c$ = 0.3) | 162 | 30.7 | 65.5 | 64.6 |

Additionally, the variation of latency and throughput with FLOPs for varying model capacities of MoNE is depicted in Figure 6a to 6d. The plots show that latency and throughput gains scales linearly with FLOPs reductions. It is important to note that inference gains depend heavily on implementation and while a simple high-level efficient implementation of our framework yields gains of this scale, we believe that further improvements can be obtained by optimizing a low-level GPU kernel implementation for MoNE.

In addition, it is worth noting that the proposed Expert Preferred Routing (EPR) in Algorithm 1 loops only over the number of experts, which is typically a small number and fixed to 4 in our framework. While the nature of the EPR algorithm does not allow parallelization of the computation any further, the time taken by the algorithm is negligibly small as compared to the total time taken by the model. For comparison, on a V100 GPU, the EPR algorithm adds just $0.5\ ms$ to the forward propagation time of a ViT-B/16 model ($190\ ms$), accounting for $< 0.3\%$ of the total computation time.

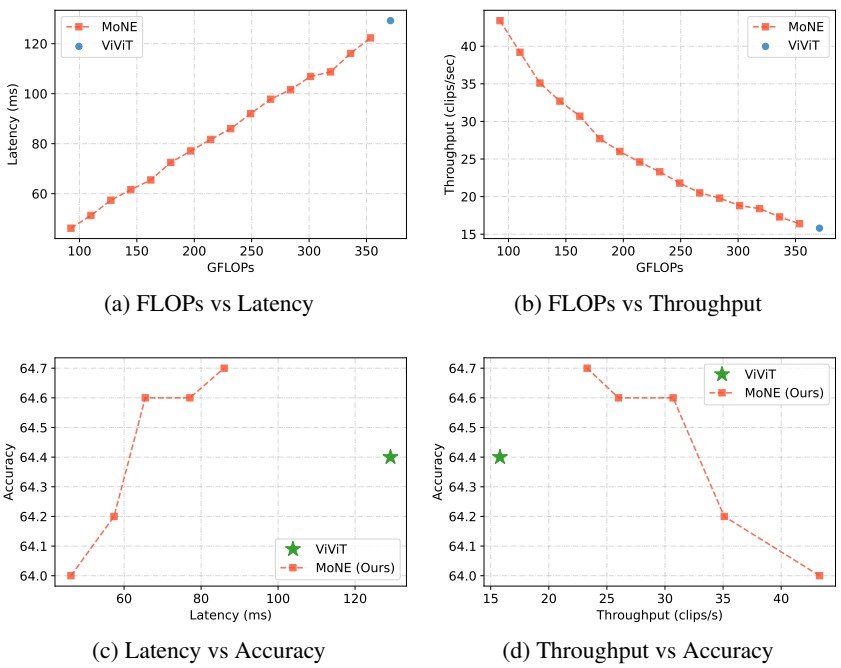

(a) FLOPs vs Latency

(b) FLOPs vs Throughput

(c) Latency vs Accuracy

(d) Throughput vs Accuracy

Figure 6: Wallclock realization of MoNE's computational savings with varying effective capacities, depicted on the Something-Something-v2 dataset.

# 6 Router Analysis

In this section, we discuss, analyse and visualise the design choices in implementing the router network. We choose the SSv2 dataset for this analysis. We further provide an extended discussion in A.3, detailing on the choice of number of routers, associating router outputs with Task Difficulty, and understanding the implications of having of learnable router.

**Router Position:** As discussed before, we use a single router at the first layer, and propagate its decisions for all layers. While a delayed router might benefit from a more processed feature representation as input, this also diminishes the compute gains, as the initial layers operate at full capacity. We reason this choice by monitoring performance while placing the router at different layers in the network. As Figure 7a suggests, the gains through richer features from the later layers is outweighed by the shift in the curve to the right, and an equivalent capacity with our default router produces higher points on the curve.

**Number of Routers:** We vary the number of routers, placing them at different regular intervals in the network in Figure 7b. The decision from one router is carried out until the next router block is encountered. We notice a clear downtrend in performance with increase in number of routers from being present in the first layer to being present in all layers. Intuitively, more routers demand learning more decisions, and the network has to adapt to these decisions, making optimization harder.

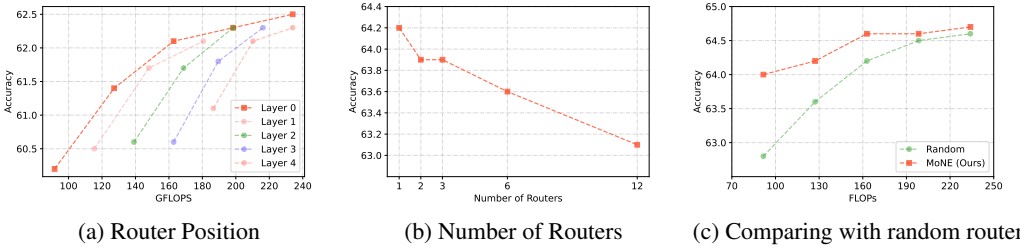

(a) Router Position

(b) Number of Routers

(c) Comparing with random router

Figure 7: **Router Analysis:** Effect of router placement and learning on Something-Something v2.

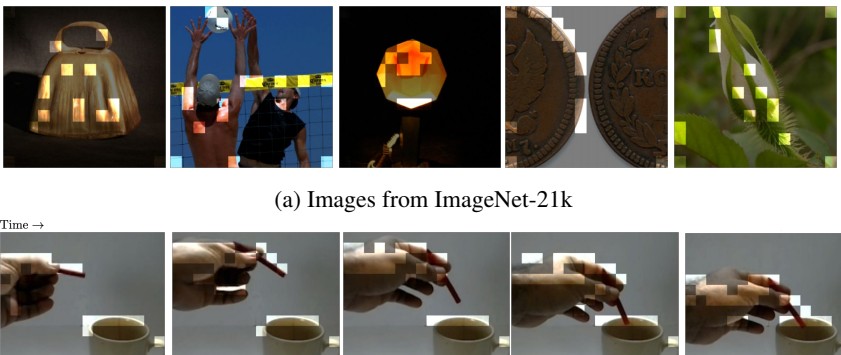

(a) Images from ImageNet-21k

Time →

(b) Video frames from SomethingSomethingv2

Figure 8: **Tokens routed to the full model:** Highlighted regions are the tokens sent to the full model, while rest of the tokens are sent to the smaller nested models. (a) shows examples on images and (b) shows an example on a video at multiple temporal indices. As we can see, the necessary and important tokens are sent to the full model.

**Comparison with Random Router:** We compare our learned router approach to a random router, which maps tokens to nested experts randomly, while still maintaining the capacity limits of each expert ($c_i$), as computed in Section 4.3. Results in Figure 7c suggests that with lower effective capacities, the random router performance degrades while the learned router still manages to understand relevant patterns from the input, thus upholding performance.

**Visualizing Important Tokens:** The above claim is further backed by visualizing the token importance during inference at a low effective capacity ($e_c$). We highlight the tokens selected by the largest expert, i.e., the full model on a few images in Figure 8a. It can be easily observed that the tokens sent to the largest model correlate well with the regions of interest in the images. On videos (Figure 8b) as well, the highlighted regions across temporal stamps consistently track the regions of motion.

**Capacity Allotment:** Given a fixed input capacity $e_c$, we demonstrate the superior performance of our heuristic-based allocation method (Section 4.3) compared to other approaches, as shown in Table 2. While the Proportionate allocation (assigning capacity inversely proportional to expert compute cost) and Uniform allocation (assigning equal capacity to all experts) show promising results, they lack the flexibility to adapt to varying budgets. Additionally, greedy approaches, such as allocating the entire budget to the largest expert and dropping other tokens (MoD style), or a greedy approach where the largest expert is assigned capacity such that all the remaining tokens are routed through the smallest expert, exhibit inferior performance.

Table 2: SSv2 Performance of different capacity distribution methods

| | Static budget | | Dynamic budget | | | |
|---|---|---|---|---|---|---|
| Distribution | Proportionate | Uniform | MoD Greedy [37] | Greedy | MoNE | MoNE |
| Effective Capacity ($e_c$) | 0.27 | 0.47 | 0.4 | 0.4 | 0.3 | 0.4 |
| Accuracy | 64.3 | 64.6 | 63.9 | 64.2 | 64.2 | **64.6** |

## 7 Conclusion

In this work, we presented Mixture of Nested Experts (MoNE), a novel framework for adaptive processing of visual tokens by dynamically allocating computational resources to different tokens. Through a nested structure with shared parameters and the proposed expert-choice routing algorithm, MoNE achieves significant reductions in inference time (over two-fold) without sacrificing accuracy on benchmark image and video datasets. Future works can be centered around extending MoNE to denser tasks like object detection, captioning, etc.

**Limitations:** Extending this to auto-regressive decoding in LLMs is non-trivial, as this is designed primarily with an encoder architecture in mind. We leave this further exploration for future work.
**Societal Impact:** The MoNE framework dynamically allocates computational resources with a given budget, thereby significantly minimizing energy usage and carbon emissions during inference of vision models. MoNE can also play a role in democratization of AI, allowing broader access to trained models without the need for large resources.

## Acknowledgements

We are grateful to Debapriya Tula, Jeevesh Juneja, Matthew Wallingford, Pradeep Shenoy and Ali Farhadi for helpful discussions and feedback.

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

# A  Appendix

## A.1  MatFormer Structure on Model Dimension

Following MatFormer convention, we define $E$ ViT blocks $\mathcal{B}_i$, such that $\mathcal{B}_i \subset \mathcal{B}_{i+1}$ for all $i \in [E]$, meaning that the parameters of $\mathcal{B}_i$ are contained in those of $\mathcal{B}_{i+1}$. With $d_i$ denoting the hidden dimension corresponding to nested model $\mathcal{B}_i$ such that $d_1 < d_2 < \dots d_E = D$, the block operation for a nesting $\mathcal{B}_i$ on an input token set $\mathbf{X} = \{\mathbf{x}_i\}_{i=1}^N$ for $\mathbf{x}_i \in \mathbb{R}^D$ is given by:

$$\mathcal{B}_i(\mathbf{X}) \triangleq \mathcal{B}(\mathbf{x}, \mathbf{d_i}) = \mathcal{B}^{\mathrm{FFN}}(\mathbf{Z}, \mathbf{d_i}), \quad \mathbf{Z} = \mathcal{B}^{\mathrm{SA}}(\mathbf{X}, \mathbf{d_i}) + \mathbf{X}, \qquad \mathbf{d_i} = \underbrace{(d_i, d_i, \dots, d_i)}_{N \text{ times}} \tag{4}$$

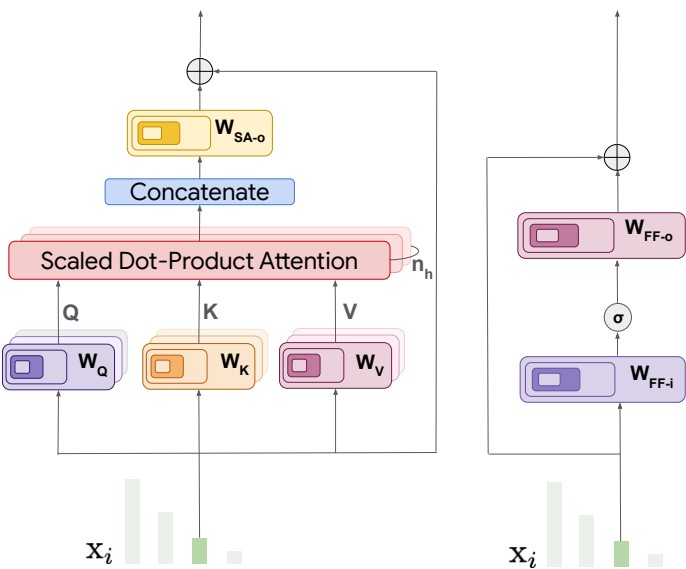

Figure 9: MatFormer Structure on Model Dimension

The modified Self-Attention $\mathcal{B}^{\mathrm{SA}}$ and Feed-Forward $\mathcal{B}^{\mathrm{FFN}}$ subroutines are shown below.

$$\mathcal{B}^{\mathrm{SA}}(\mathbf{X}, \mathbf{d}) = \mathrm{LN}\left[ \left( \sigma\left( \frac{(\mathbf{X} \odot_{\mathbf{d}} \mathbf{W_Q}) \cdot (\mathbf{X} \odot_{\mathbf{d}} \mathbf{W_K})^T}{\sqrt{d_m}} \right) (\mathbf{X} \odot_{\mathbf{d}} \mathbf{W_V}) \right) \boxdot_{\mathbf{d}} \mathbf{W_{SA_o}} \right] \tag{5}$$

$$\mathcal{B}^{\mathrm{FFN}}(\mathbf{X}, \mathbf{d}) = \mathrm{LN}\left[ \sigma(\mathbf{X} \odot_{\mathbf{d}} \mathbf{W_{FF_i}}) \boxdot_{\mathbf{d}} \mathbf{W_{FF_o}} \right] \tag{6}$$

where $\odot_{\mathbf{d}}$ and $\boxdot_{\mathbf{d}}$ respectively denote the sliced in and out projection operators, such that:

$$(\mathbf{X} \odot_{\mathbf{d}} \mathbf{W})_j = (\mathbf{x}_j)_{[:d^j]} \cdot \mathbf{W}_{[:d^j]} \qquad (\mathbf{X} \boxdot_{\mathbf{d}} \mathbf{W})_j = \mathbf{x}_j \cdot \left(\mathbf{W}_{[:d^j]}\right)^T \tag{7}$$

In the general Mixture of Nested Experts (MoNE) setting discussed in Section 4.1, the overall block computation for the set of tokens $\mathbf{X}$ requires knowledge of the expert assignments for each token beforehand. Given these assignments $\mathbf{m} \in \mathbb{R}^N$, such that $m_i \in \{1, 2, \dots, E\}$, the computation for the $i^{th}$ token processed by the $j^{th}$ expert can be represented as:

$$\mathcal{B}_j(\mathbf{x}_i) \triangleq [\mathcal{B}(\mathbf{X}, \mathbf{d})]_i, \qquad \mathbf{d} = (d_{m_i})_{i=1}^N \tag{8}$$

In Eq. 8, the block update for token $\mathbf{x}_i$ is dependent on the complete input set $\mathbf{X}$ and their respective expert assignments $\mathbf{m}$, but we omit these in the definition $\mathcal{B}_j$ for notational convenience. Additionally,

Table 3: MoNE Comparison on ImageNet-1K with other Adaptive Baselines

| Method | FLOPs (G) | Throughput (clips/sec) | Top-1 Accuracy |
|---|---|---|---|
| ViT [20] | 1.3 | 3410 | 71.3 |
| PonderNet [3, 21] | 1.0 | - | 66.2 |
| DepthAdapt [21] | 1.1 | - | 69.4 |
| ACT [25] | 1.0 | - | 71.0 |
| A-ViT [52] | 0.8 | - | 71.0 |
| MoNE (Ours) | **0.8** | **4333** | **71.4** |

this definition directly extends to the sub-routines $\mathcal{B}^{SA}$ and $\mathcal{B}^{FFN}$, as presented in Eq. 1. Here, the weight matrices of SA are $\mathbf{W_Q}, \mathbf{W_K}, \mathbf{W_V}, \mathbf{W_{SA_o}} \in \mathbb{R}^{D \times ((D/n_h) \times n_h)}$ and the weight matrices of FFN are $\mathbf{W_{FF_i}}, \mathbf{W_{FF_o}} \in \mathbb{R}^{D \times d_{ff}}$, ignoring bias terms for simplicity. $\mathbf{W}_{[:k]}$ denotes the first $k$ rows of $\mathbf{W}$. Here, $n_h$ denote the number of heads in the attention mechanism, $d_{ff}$ denotes the feed forward dimension, and $\sigma$ denotes a non-linearity, typically set to GeLU [26].

## A.2 Comparing MoNE with Other Adaptive Baselines

Firstly, we establish that MoNE, complements rather than competes, with the traditional Mixture of Experts framework. Traditional MoEs like Sparse VMoE [39] route inputs in each layer to one out of $k$ independent experts (typically the FFN block), each having the same parameter footprint, thus increasing the parameter space k-fold for the expert blocks. On the other hand, independent MoNE do not increase the parameter space, and thus MoNE blocks can potentially be used as experts in the MoE framework.

MoNE acts as an in-place replacement for a dense model like ViT, hence all our comparisons maintain the same parameter space. VMoE frameworks show cross-scale results at the expense of increased parameter space (e.g., equivalent performance of VMoE-L/16 to ViT-H/14 in Table 2 in [39], and similar cross-scale comparisons in Figs. 4 to 8 in [1]). MoNE, in contrast, matches baseline performance with limited inference compute while working with the same parameter space.

While generally MoE architectures are designed with the expectation of specialization of experts to certain tasks, this is not always the case. In Mixtral of Experts [30], the authors do not observe patterns in the assignment of experts based on the topic. In Sparse VMoE [39], the authors observe very weak correlation of router decisions to categories. In MoNE, overlap between experts allows the largest expert to utilize the full parameter space, meaning complete representation power as enjoyed by the equally-sized vanilla model. Additionally, as shown in Table 5 of [19], joint optimization of shared experts leads to better performance than having independent experts of the same size.

In Table 3, we compare MoNE with other baselines, particularly with adaptive computation of dense models. We perform this experiment on ImageNet-1K with a Ti/16 sized model. ACT [25], PonderNet [3], DepthAdapt [21], A-ViT [52] are works with similar motivation of input adaptivity as MoNE, and MoNE shows superior performance. Latency gains on bigger models e.g., ViT-B are even higher, as also observed in literature [8].

We also highlight that MoNE can be utilized as a baseline for further inference-time optimizations for improving latency. To this end, we apply Token Merging (ToME) [8] on top of the MoNE style ViT-Ti/16 model trained on ImageNet-1K. For this experiment, we train a model with full capacity till the third layer and

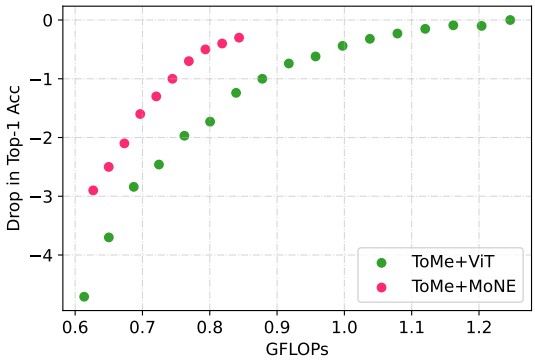

Figure 10: Latency gains by using ToMe on top of MoNE, compared against ToMe over vanilla ViT. We use a ViT-Ti/16 model trained on ImageNet-1K for this comparison.

then place a router that optimizes for latency for all subsequent layers. ToMe is applied only on the first 3 layers.

For fair comparison, we compare the performance drop and quote the same from a ViT-Ti model from the ToMe paper in Figure 10. Our preliminary results demonstrate that this implementation improves performance compared to ToMe on ViT, and this can be further extended to all MoNE layers, applying it to distinct sets of nested tokens, indicating that ToMe is complementary to MoNE.

### A.3  Understanding the Router Behaviour

**Number of Routers**: It is important to note that the number of routers in MoNE do not have the same implications as in traditional MoE frameworks. In MoEs, the parameter count increases with the number of layers on which the expert router is placed, and hence we typically see performance gains. Even then, as depicted by Tables 2 and 8 in Sparse VMoE [39], significantly increasing the parameter count with more routers only marginally improves performance. On the other hand, in MoNE, the parameter size remains fixed irrespective of the number of routers: the only change a router brings is re-assignment of tokens to nested experts while keeping the total compute per layer fixed. We hypothesize that increasing the number of routers leads to slight decrease in performance, as seen in Table 7a due to two reasons:

- It brings in additional optimization challenges (also prevalent in the MoE literature [39])
- Reassignment of a token from smaller to larger nested expert limits its information content to the representation power of the smaller expert, therefore not improving performance. The converse case occurs while reassigning from bigger to smaller nested experts, thus losing information.

Since MoNE allows flexibility in the placement of routers, an interesting future direction would be to extend MoNE to more challenging task settings, where a higher number of routers might lead to better results.

**Task Difficulty**: To further analyse the decisions made by the MoNE router, we study the visual inputs from the ImageNet-21K dataset deemed most and least compute intensive with respect to the router logits. This analysis experiment is performed in a setting without capacity constraints, in order to understand if the router decisions correlate with task difficulty (i.e. harder to understand inputs are sent to larger experts). Therefore, instead of using the greedy EPR Algorithm 1, we take an $argmax$ over the router logits to make decisions. The results presented in Fig. 11 depict two sets of images, the top-3 images that demand the lowest and the highest compute respectively, according to the router decisions. It can be intuitively observed that the images demanding less compute are visually simple, while the ones demanding highest compute are relatively complex.

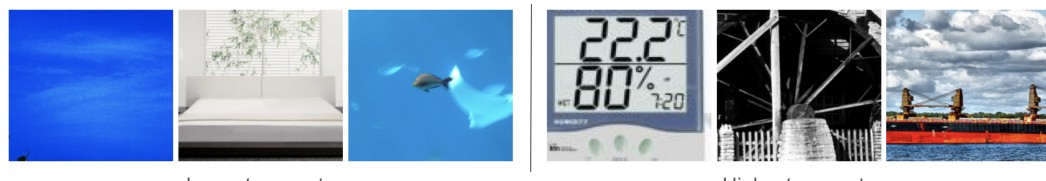

Lowest compute    Highest compute

Figure 11: A few images, which have been predicted by the router to require lowest and highest computation, from over 2000 randomly selected images in the ImageNet21k dataset.

**Learned vs Random Router**: Figure 7c shows the performance of the model with a learned vs a random router at different capacities. While for higher capacities, the learned router performs marginally better than the random one, the gap significantly widens as we go to lower capacities, from $0.1\%$ at $e_c = 0.6$ to $1.3\%$ at $e_c = 0.2$. This makes sense: with ample capacity, many tokens can be heavily processed, reducing the need for smart routing. Conversely, in low-capacity scenarios, routing decisions become crucial as only a few tokens can utilize the heavy experts. Interestingly, ViTs inherently shuffle information [14], potentially even in the "Random" router setting as well, acting as an intrinsic information router. We note that a model trained with a learned router when evaluated with a random router, performs significantly worse (~ $6\%$ drop in Top1 Accuracy on Ti/16 trained on ImageNet-1K).

