# OpenReview forum: "Mixture of Nested Experts: Adaptive Processing of Visual Tokens"
_NeurIPS.cc/2024/Conference — NeurIPS 2024 poster_

### Official Review · Reviewer_R1sX · 2024-07-10

**Soundness:** 3
**Presentation:** 3
**Contribution:** 2
**Rating:** 4
**Confidence:** 4

**Summary:**

The paper builds on the Mixture-of-Experts paradigm for vision transformer, and adds a hierarchical aspect to it, yielding a Mixture of Nested Experts (**MoNE**). More specifically, experts are defined as subsets of nested channels in the Feed Forward Networks, such that each expert has a different compute cost, and the larger one correspond to the full original model. Nevertheless, some operations are always performed at the full model dimension, after padding the tokens that went through smaller experts as needed: The $(QK^T)V$ operation in the self-attention, the layer norms and the residual connections. In addition, a dynamic budget allocation heuristic is proposed to adapted the classical (uniform) load balancing loss to the scenarios of experts with different capacities.
The proposed method is then evaluated on image and video classification.

**Strengths:**

- The routing strategy is learned in the first transformer layer and propagated to all subsequent layers. This can be better from a hardware perspective to know early on which channels need to be turned on/off.
- Having experts with different compute costs/"capacity" is well motivated and reasonable to improve the accuracy/efficiency trade-off.
- Good ablation experiments on the location of the router

**Weaknesses:**

* **Fixed number of experts:** It is not clear to me why `MoNE` uses a fixed number of nested blocks: It is very natural for `MoEs`, since experts are entirely separate blocks, however in the case of `MoNE`, the router could act directly on the channel and output a binary decision (essentially, having as many experts as dimension $D$). I would expect this to have an impact on the load balancing loss, but it would given much more flexibility to distribute the capacity across experts. More generally, there is a large literature on dynamic sparsity (e.g. *(1)* for recent references) which seems closely related to the design of nested experts and is not mentioned here.

* **Baselines:**  The experiment sections lacks baselines, in particular ones which also consider a for dynamic routing (e.g. there are none for the video classification task). For instance:
	  * Simple mixture of experts. The only baseline considered in the paper is MoD. Not only that the paper compared to *the best reported MoD configuration* (line 233); However MoD experiments seem to have been run on very different datasets, so it is not clear why the best configuration would transpose here.
	  * Token pruning; In particular for  image classification, there is a very large body of literature on token pruning (A-ViT, E-Vit, etc.). Since the only tasks considered in the experiments are classification, pruning tokens should be a valid dynamic computing baseline.
	In addition, some of the metrics reported in the paper could be improved:
	  * As far as I know, ImageNet-21k evaluation is not very standard as there is no official train-validation split.
	  * FLOPs are not enough to show the method's efficiency and they should be accompanied by real latency. For instance, they do not take into account extra padding operations.

* **Relevance of the dynamic budget allocation:** Table 1 shows that a simple uniform allocation for the load balancing loss performs as well as the heuristic proposed in Section 4.3. Therefore it is not clear to me whether the proposed dynamic budget allocation really has a significant impact



#### references
  * (1) Deja Vu: Contextual Sparsity for Efficient LLMs at Inference Time, Liu et al, 2023

**Questions:**

* In Equation (1), is $r_i$ in fact computed on $x_i$ (the input of the block) or on $z_i$ (the input of the FFN / output of the self-attention) ?
* I assume the padding to the dimension $D$ is necessary to preserve a common dimensionality for the tokens inside a batch ? or is there another motivation behind it ?

**Limitations:**

There is  little discussion about some limitations e.g. (i) why rely on a fixed number of experts or (ii) what are the real latency of the method. The only limitation mentioned is that it would be hard to adapt this scheme to decoder-only LLMs, which seems a bit out of scope with the paper itself.

---

> ### Author Rebuttal · Authors · 2024-08-07
>
> We thank the reviewer for their insightful comments. We appreciate the recognition of MoNE to have hardware-efficient routing design, the use of compute-aware experts to optimize the accuracy/efficiency trade-off, and the thorough ablation study demonstrating the impact of router placement. Below we answer some of the questions that the reviewer raised.
>
> **Fixed number of experts**
>
> This is a very astute point. Since the MatViT (MatFormer [16]) experiments suggest that the nested dimensionalities interpolate smoothly, it is fair to assume that MoNE is in-fact not training just a fixed number of experts (set to 4), but everything between the smallest and the largest experts. MoNE uses a fixed number of nested blocks to develop a tractable framework for capacity allocation and routing. As the reviewer mentioned, there is nothing stopping from extending this to all the interpolated dimensionalities (D) of nested experts barring the added complexity of precise capacity allocation and load balancing, which is generally non-trivial. The design choice to route between 4 models also relies on potential better serving capabilities given the individual block sizes. In short, yes, we agree that we can do MoNE with all the intermediate nested experts (not just the 4) but prototyped with the current design choice to show the benefits to begin with.
>
> We thank the reviewer for pointing to the dynamic sparsity work, we shall include them in the revised version of the manuscript. Compared to dynamic sparsity, MoNE and MatViT may offer a structural advantage, potentially leading to improved system efficiency by organizing sparsity in a more organized manner. As evident in the **Throughput/Latency** section in the Global Author Rebuttal, even with very high level implementation, the FLOP gains directly translate to throughput and latency gains.
>
> **Strengthening experimental evaluation with dynamic routing baselines and latency**
>
> MoNE, while being a MoE-style framework, differs starkly from the concept behind MoE networks. MoNE does not increase the number of parameters, unlike traditional MoE networks. In contrast, MoNE reduces computation while keeping the same parameter space. Given the constant parameter space, it allows for complementary use of other MoE methods together. We discuss this point further in the **MoE Comparison** section in the Global Author Rebuttal.
>
> We compare against MoD because the method is similar in spirit to ours, conditionally computing the inputs based on complexity. We extensively experimented with MoD and found that the settings mentioned in MoD translate best to these vision datasets as well.
>
> The reviewer correctly points out that for classification, dynamic routing and token pruning methods are valid benchmarks. We therefore compare against some of these methods, and report results in the **Baseline Comparisons** section in the Global Author Rebuttal. This shows that MoNE can offer superior performance compared to other dynamic routing methods.
>
> We benchmark our method on ImageNet-21K for images to depict the large-scale efficacy of our framework. We would like to point out that ImageNet-21K has been extensively benchmarked in the AugReg Paper [38]. Additionally, we show results on ImageNet-1K as well compared to other efficient architectures for smaller scale ViT models in the **Baseline Comparisons** section in Global Author Rebuttal.
>
> The reviewer correctly points out that FLOPs gains may not always translate to latency/throughput gains. Therefore, we present latency and throughput results as compared to baseline models in the **Throughput/Latency** section of the Global Author Rebuttal. We also present the variation of latency vs FLOPs, at different model capacities and observe that latency gains scale close to linear to FLOP gains. Also note that the”pad” mentioned in Figure 2a do not actually result in any additional computation, and can be avoided in implementation.
>
> **Significance of dynamic budget allocation**
>
> It is true that the uniform allocation performs as well as our heuristic method, but at a higher capacity requirement. More importantly, a uniform allocation $c_i=\frac{1}{4}, \forall i$ leads to a model with a fixed capacity based on the individual compute requirements of the experts, i.e., $e_c=\frac{1}{4}(\frac{1}{8}+\frac{1}{4}+\frac{1}{2}+1)=0.47$. In contrast, the proposed heuristic
> allows for flexible choice of model capacity ($e_c$) as per compute requirements. As discussed in the paper, given a user specified $e_c$, many solutions exist for $c_i$, and it is non-trivial to choose one over the other. The heuristics add constraints to reach a solution which offers promising results. The high performance of MoNE across capacities is depicted in Figures 3 and 4.
>
> **Clarification on input for computing r_i in Eqn (1)**
>
> The router logits $r_i$ are computed on the block input $x_i$, because unlike traditional MoE architectures, MoNE leverages the nested structure not just in the FFN layer, but also in the self-attention, to further increase compute gains. The exact changes to these operations are briefly described in the Appendix A.1 (Eqn. 5 and 6).
>
> **Padding to dimension D**
>
> Yes, padding is only to make sure two different dimensional vectors can be added. However, in an efficient implementation, padding would not be necessary, since we can maintain features of different dimensions separately, and add only the non-zero regions whenever needed.
>
> **Limitations**
>
> We have addressed the limitations mentioned by the reviewer in this discussion and in the Global Author Rebuttal. We will include a more elaborate discussion of our method and areas for future work in our final revision.

---

> > ### Comment · Reviewer_R1sX · 2024-08-13
> >
> > Thanks a lot for your response. While I appreciate the added throughput results and discussion, I am inclined to keep my rating as I still think this would warrant a more extensive evaluation/discussion/modification of the paper.  In the current state, the paper proposes a technically sound dynamic computing method but does not discuss/compare well to existing related work (e.g. no throughput for dynamic pruning methods), and the core evaluation in the paper is performed on very specific settings/benchmarks (imagenet21k for classification, and video classification), so it is also hard to get an insight on how it compared to existing dynamic methods.
> >
> > > We benchmark our method on ImageNet-21K for images to depict the large-scale efficacy of our framework. We would like to point out that ImageNet-21K has been extensively benchmarked in the AugReg Paper [38]
> >
> > While I do agree that ImageNet-21k might be a more robust benchmark than ImageNet-1k, it is much less standard in practice.  With respect to existing baselines, it would be more fair to report ImageNet-21k results *in addition* to 1k rather than omit the 1k benchmark.
> >
> > > MoNE, while being a MoE-style framework, differs starkly from the concept behind MoE networks. MoNE does not increase the number of parameters, unlike traditional MoE networks.
> >
> > I agree but one may argue that this is the strength of MoE: Increased model capacity / number of training parameters while keeping the number of inference parameters constant (at least for a batch size of 1). It would be easy to design a MoE  counterpart that would have the same number of inference parameters as MoNE, or inversely, the same number of training parameters but faster inference.
> >
> >
> > > However, in an efficient implementation, padding would not be necessary, since we can maintain features of different dimensions separately, and add only the non-zero regions whenever needed.
> >
> > I am not 100% convinced it would be so trivial to reach such an efficient implementation (hardware wise), as it would also mean having to constantly handle tokens with different number of dimensions, as opposed to the standard "large tensor multiplication/sums". In that sense do see the advantage of simplicity for padding, but it seems to be a bit of a negative point from a memory usage perspective (though MoE also have a similar issue with batched inference and having to activate many experts)

---

> > > ### Author Response · Authors · 2024-08-13
> > >
> > > We would like to clarify some potential misunderstandings by the reviewer, and mention that many of these points are addressed in our [global rebuttal](https://openreview.net/forum?id=HbV5vRJMOY&noteId=m6HR8cjvnf) to all reviewers.
> > >
> > > In particular, we have already compared to existing dynamic models on ImageNet-1K in our initial rebuttal. Moreover, our current implementation does not use unnecessary padding, and we have already demonstrated substantial gains in latency (in addition to theoretical FLOPs) in our initial rebuttal. Finally, we also discussed in our initial rebuttal about how our MoNE is complementary to traditional MoEs.
> > >
> > > We now detail these points below:
> > >
> > > **Baselines**
> > >
> > > We compared to existing dynamic computing methods in the second table of our initial [Global Rebuttal](https://openreview.net/forum?id=HbV5vRJMOY&noteId=m6HR8cjvnf). This table shows that we outperform existing approaches in both accuracy and GFLOPs when using the same ViT-Ti backbone. Note that throughput comparisons are difficult as they are hardware and implementation dependent (eg. ViT-B/16 throughput mentioned in [47] is 300 img/sec vs in [38] is 659 img/sec), thus we have only compared our method to a vanilla ViT using the same hardware and code-base.
> > >
> > > **ImageNet-1K**
> > >
> > > To clarify, our **Baseline Comparisons** table in the [Global Author Rebuttal](https://openreview.net/forum?id=HbV5vRJMOY&noteId=m6HR8cjvnf) is indeed on the **standard ImageNet-1k benchmark**. It indicates the superior performance at lower FLOPs of over other dynamic methods _in addition to_ the extensive ImageNet-21K results shown in Figures 3 and 5 of the paper. Additionally, the same section mentions that Token Merging techniques (ToMe) can be applied on top of MoNE to further achieve latency savings, thus depicting our framework’s flexibility of use and how it is complementary to related work. The performance of MoNE is further shown on **standard** video classification benchmarks (Kinetics400, SSv2), indicating that MoNE is able to achieve baseline performance at significantly lower performance. We hope these clarifications help and are happy to clarify further.
> > >
> > > **MoEs**
> > >
> > > As detailed in the [Global Author Rebuttal](https://openreview.net/forum?id=HbV5vRJMOY&noteId=v0FuPA3Sk5), MoNE is comparable to a dense model, as it does not increase the parameter space, and we have demonstrated accuracy and efficiency gains over dense models. Extending MoNE to a sparse MoE setup is a promising direction for future work.
> > >
> > > Concretely, we can use MoNE in a sparse MoE setup where the experts are multiple MoNE layers, instead of traditional dense layers. This would reduce the compute cost of an MoE whilst still having the same parameter space.
> > >
> > > In response to "It would be easy to design a MoE counterpart that would have the same number of inference parameters as MoNE …", it is important to note that the parameter-scaling properties of MoE models are different to dense models, particularly in the "low-parameter" regime. As shown in Table 8 of VMoE [34], MoE B/16 ,models that have the same number of parameters as dense ViT L/16 models underperform the dense models by 2-5% depending on the task.
> > >
> > > **Efficient Implementation**
> > >
> > > We would like to clarify that while Figure 2a of the paper indicates a padding for bringing feature representations to the same dimension, it is  _only_ for illustration / ease of explanation. In our actual efficient implementation, we handle tokens with different dimensions as different tensors, processing them in parallel without padding, and concatenating only for combined operations that require the whole token set together, which happens to be the case only for the SoftMax operation in SelfAttention. In this way, we match the exact theoretical FLOPs as well as achieve the gains indicated in the **Latency/Throughput** section of the [Global Author Rebuttal](https://openreview.net/forum?id=HbV5vRJMOY&noteId=v0FuPA3Sk5). Consequently, from a memory perspective as well, we do not incur any additional costs as we perform the exact operations _without padding_.

---

### Official Review · Reviewer_LDxY · 2024-07-12

**Soundness:** 3
**Presentation:** 4
**Contribution:** 3
**Rating:** 7
**Confidence:** 4

**Summary:**

This paper present a method to select nested portions of a transformer network, using a MoE router-expert assignment method where each expert is a progressively larger slicing of a single underlying model.  A capacity budget determines how many tokens can go to each expert, while a router network scores experts for each token so that the most important tokens (in the sense of benefitting from the larger computations) go to the larger model slices.  Furthermore, when trained using random budget selection, the model can be dynamically scaled to different cost-accuracy tradeoffs at inference time.  The model is evaluated on image and video classification tasks using imnet21k, K400 and SSv2, with excellent flops-accuracy performance compared to baselines, and a set of ablations show the impact of different design choices for router placement.

**Strengths:**

This is a well-written paper that describes an in interesting and effective idea.  The approach is evaluated convincingly on three datasets for image and video classification tasks, and in enough settings to profile its performance characteristics and anecdotal visualizations of its behavior.

**Weaknesses:**

There are a few points that I think were under-explained (see questions below).  In particular the descriptions of the projections around the MLP and SA were a little terse, and I'm still not sure exactly which operations are in the subspaces vs the full dimensional space.

The connection between token redundancy and the operation of the model is also a little tenuous, though likely (see below as well), and while FLOPs are measured explicitly, the impact on both real computation savings and runtime may depend on hardware and distributed computation implementations, which I didn't see mentioned.

**Questions:**

* sec 3.1  I don't understand exactly where the projection back out to 4*D or D happens, and if this was a linear projection or padding?  What operations are performed in the full-dimensional space, and which in the subspaces?  Right now it's unclear exactly where the computation savings are, as I'm not sure exactly which operations are in the smaller subspace.

* sec 3.1 l.102:  "it is always projected to the model dimension D for the (QK)V operation."  This could use more explanation on how the projection is done.  Is this linear or padding?  A more explicit equation for the projection from D/m to D could help.

* Which tensors and dimensions are the layernorms performed over?  It appears they are performed within each expert separately (which would make sense since they have different dimensions) but I'm not entirely sure.

* The motivation mentions redundancy in the tokens quite a bit, and while I agree with that in general, I think the link so far is tenuous and wasn't made very explicit.  In particular, if the router network is a linear classifier on the first transformer input, then how can it possibly know which tokens are redundant without comparing between them?  Does such a comparison come from lower conv layers in feature extraction?  Or is it just that some types of tokens tend to be more redundant than others (flat regions, for example, are just by nature of being flat and therefore next to something similar).

* eq 3:  the effective capacity e_c is defined as a linear combination of the dimension ratios, but d-dimensional MLP will use O(d^2) operations, so that a d/2 dimensional MLP will have 1/4 of the multiply-adds compared to a d dimensional one.  Does this matter for the capacity assignments?  Or is this not actually the case because the MLP always has one 4D hidden layer, so it is a linear scaling?

* also eq 3:  the overall intuition makes sense as described, but this particular optimization seems a little arbitrary; in particular, why are the weights delta^i exponential, and why is the entropy term needed in addition to the e_c capacity constraint?  It seems possible the entropy is counteracting the exponential delta weights, when this could have been less aggressively weighted with no entropy?  I don't know that there is necessarily a benefit to equal usage of all experts in this case, either (see my other question on runtime and distributed computation below)

* for each of the different FLOPs levels in the performance curves, what are the expert assignment ratios?

* Another interesting point of comparison would be the accuracy levels of each expert alone --- that is, the 4 single-expert routing assignments (1,0,0,0), (0,1,0,0) etc.

* an interesting extension would be a lowest-capacity expert with 0 in its dimension slice, so no MLP, but still included in self-attention as well.  this one is perhaps beyond the scope of this work but seems natural considering the redundant computation motivations

* something that wasn't touched on was the impact on model runtime and latency, particularly in distributed settings, although that will be hardware-dependent.  Since there are sync points between the experts at all SA layers, how to distribute computation between devices isn't as immediately clear as for many same-capacity experts, since the longest-running computation can be in any of the four expert capacities depending on the assignments

* sec 4.4 videos:  I agree that there is redundant temporal information and that would be a great application of MoNE.  However, the application in the second paragraph doesn't actually seem to leverage or implement that, since it says that it applies the MoNE frame-by-frame.  If that's the case, the temporal dimension wouldn't be used in MoNE to determine important vs redundant computations.  If time and space combinations interleave or alternate, then it could --- but the description indicates that isn't the case.  On the other hand, even without this just looking at a single frame, some regions are more likely to be redundant across frames than others, and so the argument for MoNE may still apply because of that.  Whatever the case, it appears contradictory in the text right now and could use a little more explanation.



Smaller comments:


* l.122:  "with k = 1" isn't very clear, I initially thought it meant "if k = 1" which prompted me to ask, but what if k > 1?  In fact, I think this means to say e_i is a subset of e_i+1 and k is always 1 (i.e. exactly one of the nesting levels is selected)

* meaning of i, j isn't always consistent, e.g. l.114 and 168 sums i over 1..E, but in most other places i is tokens and j is experts, so seeing i index the experts in a few places is a little confusing

* Alg 1 EPR:  Should "T" be "N" here?  Also, the way this is phrased with floor function, there could be a few left-over tokens at the end that are unassigned.  I take it these are assigned to the cheapest expert rather than being dropped?

* l.185:  "flexibility" --- If the flexibility here is dynamic and can be specified at inference time, then c would need to be changed at inference time as well --- does this mean that many different values for c are used a training time?  And how are they randomized or selected?   --- this was answered in adaptive training experiment later in sec 5, but could be good to mention in the method description

**Limitations:**

addressed in the final discussion section

---

> ### Author Rebuttal · Authors · 2024-08-07
>
> We thank the reviewer for the positive feedback and glad to see that the reviewer found our work to be well-written, interesting and effective with comprehensive eval.
>
> **Projections and compute save. Projection from D/m to D for (QK)V.**
>
> In a Transformer model, there are total 8 primary linear projection operations -
> (1) 3 to obtain $Q=W_qx$, $K=W_kx$, $V=W_vx$,
> (2) 2 to obtain $y=(QK^T)V$,
> (3) 1 in applying $z = W_oy$,
> (4) 2 in MLP: $W_2\sigma(W_1x)$
>
> MoNE saves compute in all of these ops, except (2). In transformers, the weight matrices are of size $\mathbb{R}^{d_a\times d_b}$, where $d_a, d_b$ represent the feature dim (model dim, MLP dim, etc). In MoNE, these are either $\mathbb{R}^{d_a\times \frac{d_b}{m}}$ or $\mathbb{R}^{\frac{d_a}{m}\times d_b}$, depending on the op. $m$ denotes the reduction in the feature dim for the nested expert. Hence, in all projections, nested experts have FLOP gains by a factor of $m(>1)$. Denoting the model dimension as $D$, the features are extracted from $D$ to $\frac{D}{m}$. For eg, for the nested model with dim $\frac{D}{m}$, in (1) the input $x$ is $\frac{D}{m}$ which is projected to $D$. For that, we slice a weight matrix of dim $\mathbb{R}^{D\times \frac{D}{m}}$ from the full matrix of $\mathbb{R}^{D\times D}$, and multiply it with $x$, as explained in Lines 93-94. Hence we save by a factor of $m$, by choosing the nested matrix instead of the full matrix, for which the input would be $D$. Similar explanations for (3) and (4). These operations are discussed in Appendix A.1.
>
> **Layer Normalization**
>
> As in Line 98, after linear projections, the features are padded to D, to add with the residual features, and LayerNorm on full D dim. In the next layer, a part of the feature $\frac{D}{m}$ is sent and weight slicing operations are applied to project, as discussed above.
>
> **Leveraging redundancy**
>
> By redundant, we mean that the info of a certain token set is redundant/less needed, given a few of important/needed ones, as measured by final accuracy for a given compute. Our router does learn to prioritize features like edges for the largest expert, confirming the reviewer's hypothesis. Such features can be learned by low level convolutional filters. Tasks which need to compress redundancy at a higher level might need a router deeper into the network. Additionally, ViTs are known [c] to shuffle info among tokens, acting as an intrinsic router, thus adapting itself given the initial router decisions.
>
> [c] Darcet et al., Vision transformers need registers, ICLR 2024
>
> **Eq. 3 relation between dim and capacity**
>
> Yes, the last line of the reviewer for this point is indeed correct. In the MLP dimension, the full model would do two projections: $D \rightarrow 4D$ and $4D \rightarrow D$ amounting to 2 * 4D * D compute. The nested model would do $\frac{D}{m} \rightarrow 4D$ and $4D \rightarrow \frac{D}{m}$, hence amounting to 2 * 4D * $\frac{D}{m}$ compute, a $m$ fold reduction.
>
> **Eq. 3 Rationale**
>
> The weights $\delta^i$ are exponential to account for the exponentially distributed accuracy differences (Fig 3, 4) between consecutive MatViT models. While this is empirical, this first term is as a proxy to accuracy obtained by a particular expert. The larger experts are thus favored more. However, in without the entropy term, this objective would lead to a greedy assignment of either the largest or the smallest expert. This will not leverage the framework’s full potential. The entropy term balances this, so that each expert is provided a significant capacity. We empirically reason some of our choices in Table 1 of paper, showing that this heuristic leads to competitive results across capacities.
>
> **FLOPs to Expert Capacity**
>
> The expert assignment ratio, $c_i,i\in[1,E]$ for expert $i$ is obtained by optimizing Eqn 3 for a given capacity $e_c$. Given FLOP needs, $e_c$ can be obtained by solving: GIVEN_FLOPS = FLOP_b + e_c * FLOP_a where FLOP_b corresponds to the model operations that are not reduced by MoNE (patch embedding, attention value ($(QK)^TV$) computation), and FLOP_a correspond to the linear projections.
>
> **Single expert accuracy**
>
> The individual MatFormer expert accuracy are in Fig 3 and 4 as MatViT/MatViViT.
>
> **Zero-capacity expert**
>
> This is close to skipping entire layers for some tokens in Mixture of Depth (MoD). We do compare with MoD in Fig 3, and find MoNE to perform better.
>
> **Runtime/latency**
>
> We agree that latency is heavily implementation dependent. Our high level Jax implementation resulted in latency gains scaling linearly to FLOP gains. We present this in the **Latency/Throughput** part of Global Author Rebuttal. The largest computations that bottleneck throughput can be resolved by efficient implementation. The ops are tensor products of different sizes, which can be decomposed into smaller sub-tensor products and efficiently parallelized (Fig 3 [14]), avoiding bottlenecks. We believe that further latency improvements can be obtained by low-level implementations.
>
> **Sec 4.4 videos**
>
> Our exploration for MoNE in videos is a first step based on the popular factorized-encoder ViViT [2] model, for which the gains in the spatial dimension are more readily realized, given that temporal information here is integrated late in the architecture so a router has 196*16 choices spatially vs. 16 temporally (over the 16 frame-wise CLS tokens). We will expand and clarify this discussion further; integrations with other space-time video architectures with more interleaved operations is an exciting direction for future work.
>
> **Smaller comments:**
>
> We thank the reviewer for noting the small comments. The reviewer's understanding is correct. We will clarify them in the paper. Yes, remaining tokens due to integer packing are not dropped but assigned to the smallest expert.

---

> > ### Comment · Reviewer_LDxY · 2024-08-12
> > **responses**
> >
> > Thanks for the responses.  I especially appreciate the new latency and throughput measurements and additional baselines as requested by a couple of the other reviewers.  I will keep my score at 7.
> >
> > The explanation of slicing above helped, as it explains better that each of the W matrices are sliced on one of their dimensions, but I still think it could be clearer.  I found it helpful to write out the dimensions of all the W matrices and go back over the transformer ops you enumerated, distinguishing between "D" and "H" dims in each:
> >
> > $W_q : [H_A, D]$,
> > $W_k : [H_A, D]$,
> > $W_v : [H_V, D]$,
> > $W_o : [D, H_V]$,
> > $W1 : [H_{MLP}, D]$,
> > $W2 : [D, H_{MLP}]$
> >
> > in which case all the the $D$ dims are sliced to $D/m$, and applied to correspondingly sliced vectors, and all the $H$ dims are left intact, if I understand correctly.
> >
> > I also now see that in Fig 2, the Pad operation is not shown in 2b, only 2a, and that the order of slicing and LN is inconsistent between 2a and 2b:  2a applies LN to full D-dim features, which makes sense, but 2b indicates that x is sliced before applying LN, in which case it's unlcear if this means x is padded back up to D and re-sliced, or if it's actually applied as in 2a, with LN before slicing.  I think this is where some of my confusion around these ops came from as well.  Lastly the horizontal blue bar of dim 4D in the MLP (and the horizontal red bar of dim D in SA) are a little confusing as well, as all the other sliced horizontal bars here are W matrices, and so it's not clear what the unlabeled bars correspond to in the figure.

---

> > > ### Author Response · Authors · 2024-08-13
> > >
> > > We thank the reviewer’s appreciation.
> > >
> > > The reviewer’s understanding is correct regarding slicing, and we will make it clear in the revised manuscript.
> > >
> > > In Figure 2b, we do not show the slicing operation in the intermediate layers and only show in Fig 2a, to avoid an overcomplicated diagram. Figure 2b is just to show the mixture of nestedness across tokens. The slicing at the input is just an indicative of that token processing at a certain nested dimension. The horizontal blue and red bars indicate the dimensions of the MLP and attention hidden dimension respectively, which are always at the full dimension ($H_A, H_V, H_{MLP}$ using the reviewer’s notations).

---

### Official Review · Reviewer_YGXi · 2024-07-13

**Soundness:** 2
**Presentation:** 2
**Contribution:** 3
**Rating:** 6
**Confidence:** 4

**Summary:**

This paper tried to use Matryoshka mechanism to assign tokens to different experts.

**Strengths:**

1. Seems like the proposed approach can learn some effective components in images, shown in visualizations .
2. The empirical performance is good compared to mavit.

**Weaknesses:**

1. Why there is not comparison with FF which owns a single scale? What I mean is, starting for MaViT, and continue finetune the model with the same amount of compute .
2. Why is imagenet accuracy so low compared to a normal ViT? Such as Figure 3.
3. Why is there no experiments on imagenet1k?

**Questions:**

1. In "we place a single router at the first transformer layer, and propagate the router decisions to all the layers.", why doing this? Why can't we learn from MoE papers, and place routers for all transformer layers?
2. How stable is router training? How do you evaluate whether router really can find a suitable compute to match the task difficulty for a specific image/video?
3. In figure 6b, do you mean that there is one router in the whole model? This is suprising.
4. Seems like random router is only a bit worse than the learned one...

**Limitations:**

1. Important Tokens: is there a way to measure it?
2. The method description part is not clear to me.
3. Authors should also consider comparing with TokenMerging series of work, since the goal is the same, which is to save compute from the token level.

---

> ### Author Rebuttal · Authors · 2024-08-07
>
> We appreciate the reviewer's positive feedback and are pleased they found our method produces informative visualizations and better results than MatViT.  We'll address their questions below.
>
> **Comparison with Single-Scale FF and Fine-tuning with Same Compute**
>
> We appreciate the reviewer's suggestion regarding fine-tuning our framework from a MatViT model with the same compute. We exactly follow this procedure for videos, as detailed in Section 5, Lines 246-254. We utilize isoFLOPs training [24, 32] across all methods in Figure 4, meaning equal training FLOPs. While the x-axis in Figures 3, 4 depict the inference FLOPs consumed by the individual expert models (MatViT and MatViViT) and MoNE models with varying capacities, they are trained in isoFLOPs manner. We'll clarify this further by updating the figures and their captions.
>
> For images, we show that MoNE can be trained from scratch for the same number of epochs as ViT (thus taking much lower training FLOPs), and still perform favorably to ViT and MatViT (Figure 3). In Figure 3a, we show that MoNE’s performance can be further improved by using the same training FLOPs as ViT (MoNE-isoFLOPs). Note that MatViT performs 4 forward passes per training step, while MoNE performs just one. Regarding comparison with a single scale, Figures 3 and 4 depict the performance of individual MatFormer models. We will highlight these points in the revision.
>
> **Low ImageNet Accuracy**
>
> The numbers in Figure 3 correspond to validation accuracy on ImageNet-21k, and they are on-par with those reported in Fig. 4 of AugReg paper [38], with ~2x reduction in compute, for all three S, B and L models.
>
> **Experiments on ImageNet1k**
>
> We primarily experiment on ImageNet-21k to showcase our model's effectiveness in large data regimes. These have been elaborately benchmarked earlier in the AugReg [38] paper. But, as ImageNet1k is a widely used benchmark for smaller scale ViT models, we compare MoNE with other adaptive ViT models in the **Baseline Comparisons** section of the Global Author Rebuttal.
>
> **Router Placement**
>
> Our method substantially differs from traditional MoE frameworks. Unlike MoE, where parameter count increases with routers in more layers, our method maintains a constant parameter count regardless of the number of routers. This methodological difference explains why MoE router settings don't directly apply to our framework.
>
> We experiment with routers placement and present results in Figures 6a,b. Experiments suggest the best setup is a single router at the first transformer layer. While having multiple routers (expert choice) is the norm in the MoE literature, the MoNE setup is different from MoE. We further extend this discussion in the **Number of Routers** section of the Global Author Rebuttal.
>
> **Router Stability and its effectiveness in identifying to task difficulty**
>
> The router is jointly trained with the model to optimize only the classification loss, and we find the training to be stable, as suggested by the high performance of the framework and relevant visualizations in the paper. Figure 1 and 7 show that MoNE can learn to assign important tokens to the higher sized models, and the redundant ones to the lower sized nested models.
>
> The question raised about task difficulty is a really interesting one. While we fix a capacity and route tokens according to importance, as would be necessary for real-time applications, we do see the model is inherently able to learn the difficulty level of a specific data point. We discuss and visualize these results in the **Task Difficulty** section of the Global Author Rebuttal. These results validate that the router is able to associate compute requirements with task difficulty.
>
> **Random vs Learned Router**
>
> Figure 6c shows the performance of the model with a learned vs a random router at different capacities. While for higher capacities, the learned router performs marginally better than the random one, the gap significantly widens as we go to lower capacities, from 0.1% at e_c = 0.6 to 1.3% at e_c = 0.2. This makes sense: with ample capacity, many tokens can be heavily processed, reducing the need for smart routing. Conversely, in low-capacity scenarios, routing decisions become crucial as only a few tokens can utilize the heavy experts. Interestingly, ViTs inherently shuffle information [c], potentially even in the "Random" router setting as well, acting as an intrinsic information router. We note that a model trained with a learned router when evaluated with a random router, performs significantly worse (~6% drop in Top1 Acc on Ti/16 trained on ImageNet1k). We will update the paper with these discussions.
>
> [c] Darcet et al., Vision transformers need registers, ICLR 2024
>
> **Measuring Token Importance**
>
> The router leads to intuitively sensible decisions on a per-token basis, as depicted in Figures 1 and 7, on both images and videos. The router decisions correlate well with the important tokens in the image/video. In Figure 7a, we see that the relevant image regions are processed by the largest expert. In Figure 7b, we see that the tokens sent to the largest expert correlate well with the motion in the video. This is a qualitative measure of token importance, and the router decision logits can thus be taken as a quantitative measure as well, which the EPR algorithm does while assigning tokens to experts.
>
> **Comparison with Token Merging**
>
> Since TokenMerging algorithms can work on top of any ViT-like models, it is complementary to our method and can potentially be applied to a pre-trained MoNE model to further reduce computation. We empirically validate this claim in **Baselines Comparisons** in the Global Author Rebuttal and compare against other adaptive algorithms as well. Note that a naive implementation already performs well and this can be further improved by considering the MoNE architecture into account, as discussed. We will add this result to the revised manuscript.

---

### Official Review · Reviewer_kK61 · 2024-07-13

**Soundness:** 3
**Presentation:** 3
**Contribution:** 3
**Rating:** 6
**Confidence:** 4

**Summary:**

This paper introduces the concept of Mixture of Nested Experts (MoNE), which utilizes a nested architecture to process visual tokens more efficiently in visual media like images and videos. MoNE aims to leverage redundancy in data, choosing experts in a priority order to process visual tokens, thereby achieving substantial compute savings while maintaining performance. The approach, demonstrated with MoNE's algorithms like MoNE-21K and MoNE-4K, optimizes adaptive processing on standard image and video datasets, significantly reducing computational demands while using a single trained model.

**Strengths:**

1. Overall, the paper is well-motivated and the idea of combining nested structers with MoE is very interesting; The problem of information redundency do exist in image or video classification tasks.

2. Empirically, the MoNE models attain very competitive results with less FLOPs and parameters, which could support their theoretical analysis in vision information redundency.

3. The paper is well-written and easy to understand. The method is simple and easy to follow.

**Weaknesses:**

1. My main concerns are in the actual inference speed of your models. The experimental results reported in the paper focus on comparing FLOPs rather than throughput. Could the authors demonstrate some direct comparisons of training/inference speed?

2. The EPR algorithm (Algorithm 1) proposed in the paper seems to implement routing operations through some loops. Is this process actually implemented through loops or parallel computing? Does this operation take up a lot of inference time?

3. I'm also concerning whether the method can benefit dense prediction tasks such as image segmentation. As in those tasks there are less information redundency, will MoNE result in performance degradation?

**Questions:**

See weaknesses.

**Limitations:**

Limitations are discussed in the paper.

---

> ### Author Rebuttal · Authors · 2024-08-07
>
> We thank the reviewer for the positive feedback. We are glad to hear that the reviewer found the core idea of the work – nested structures to exploit information redundancy – to be well motivated and interesting, the experimental results to be promising and the paper to be well-written. Below we address some of the questions raised by the reviewer.
>
> **Actual inference speed comparisons (Throughput/Latency)**
>
> We agree with the reviewer about the importance of realizing the FLOP gains in terms of latency/throughput. We provide real inference speedups in the **Latency/Throughput** section of the Global Author Rebuttal. We will include these results to the main paper.
>
> **EPR algorithm implementation**
>
> The EPR algorithm contains a loop only over the number of experts, which is quite low and fixed to 4 in our framework. While the nature of the EPR algorithm does not allow to parallelise the computation any further, the time taken by the algorithm is negligibly small as compared to the total time taken by the model. For comparison (on GPU), for a ViT B/16 that takes 190ms for forward propagation, the EPR algorithm takes 0.5ms, less than **0.3%** of the total computation time. We will mention this in the revised paper.
>
> **Applying MoNE to dense prediction tasks**
>
> Thank you for this suggestion. As dense prediction tasks typically operate on higher resolution images, we believe MoNE can offer further computational gains as the number of input tokens increases. Dense prediction tasks also provide stronger supervision at each pixel, which may help our model learn redundancy in the data. Leveraging MoNE for tackling redundancy in denser tasks would be an exciting direction for future work.

---

> > ### Comment · Reviewer_kK61 · 2024-08-13
> > **Reply to rebuttal**
> >
> > Thanks for the authors' rebuttal. Most of my concerns are well-addressed and I keep my rating of a weak accept.

---

> > > ### Author Response · Authors · 2024-08-13
> > >
> > > We're glad to hear that our rebuttal addressed your concerns.

---

### Official Review · Reviewer_E3Y8 · 2024-07-17

**Soundness:** 3
**Presentation:** 3
**Contribution:** 3
**Rating:** 6
**Confidence:** 4

**Summary:**

This paper proposes the Mixture of Nested Experts (MoNE) framework. MoNE is built on top of the MatFormer architecture which utilizes a nested architecture where smaller, less computationally expensive sub-models are nested within larger models. Similar to MatFromer, MoNE uses structured slices of the model weights (the experts) to process information hierarchically. While MatFormer focuses on obtaining many models on the optimal loss-vs-compute curve using the mix'n'match algorithm (during inference), MoNE learns to dynamically route visual tokens to an appropriate expert based on their significance and the computational budget.

The results show that MoNE achieves a favorable accuracy-efficiency trade-off curve for image and video classification tasks compared to the baselines.

**Strengths:**

- The paper is well-written and easy to follow
- The extension of the MatFormer architecture into an MoE architecture with dynamic routing seems quite intuitive. The Matformer architecture is inherently composed of nested subnetworks and enabling dynamic routing in this architecture makes sense.
- Experiments on Image and Video Classification tasks shows MoNE outperforms the baselines.

**Weaknesses:**

- Baseline comparisons:  It is conceivable that the dynamic routing approach can outperform the mix'n'match policy of MatFormer during inference as smaller sub-networks can focus on simpler/uninformative tokens while the larger experts focus on more complex/informative tokens. Nonetheless, to fully evaluate the potential of the proposed MoE framework, it would be beneficial to compare it with other MoE architectures, especially those like Sparse Vision-MoE.

- The advantage of having a nested set of experts compared to non-overlapping experts remains unclear. The qualitative results suggest that MoNE utilizes the full model for the most critical tokens, while less informative tokens are processed by smaller sub-networks. Given this, the paper could be strengthened by comparing MoNE to other dynamic token processing methods such as AdaViT [1], SVitt [2], and A-ViT [3], which have significantly enhanced computational efficiency in image and video recognition tasks by learning to dynamically skip tokens.

1. Meng, Lingchen, et al. "Adavit: Adaptive vision transformers for efficient image recognition." CVPR 2022.
2. Li, Yi, et al. "Svitt: Temporal learning of sparse video-text transformers." CVPR 2023.
3. Yin, Hongxu, et al. "A-vit: Adaptive tokens for efficient vision transformer." CVPR 2022.

- Specialization of Experts in MoE Architectures: A primary focus in designing MoE architectures is the specialization of experts to address different aspects of the data distribution. Typically, having non-overlapping diverse experts is desirable. In contrast, MoNE and MatFormer consist of partially overlapping parameters, suggesting that the implicit learning bias in these architectures may differ significantly from that in conventional MoEs. Investigating the differences between MoNE experts and conventional sparse MoE architectures could greatly enhance the reader's understanding of the proposed MoE framework's behavior.

- Worse results when increasing the number of routers: Standard MoE architectures employ MoE layers and routers in every(other) layer, enhancing the model's flexibility and expressiveness. However, it is concerning that increasing the number of routers leads to worse performance in MoNE. This suggests potential optimization challenges in this MoE model compared to standard MoEs. Notably, the best results are obtained when the router is placed at the very first layer, indicating that the dynamic decision-making for all layers is based on relatively simple cues at local feature levels. Further exploration into why the performance deteriorates with more routers compared to traditional MoEs, which typically see improved results, would be insightful.

**Questions:**

Please review the weaknesses section.

**Limitations:**

The authors mention one of the limitation of their work regarding extension to sequence modeling tasks.

---

> ### Author Rebuttal · Authors · 2024-08-07
>
> We would like to thank the reviewer for the insightful review. We are glad to hear that the reviewer found the paper to be well-written, the MoNE framework to be intuitive, and compelling experimental results. Below we answer some of the questions raised by the reviewer.
>
> **Dynamic Routing vs MatFormer and the need for MoE comparisons**
>
> We agree with the reviewer that our dynamic framework allows for more informed token routing than the static strategy in mix’n’match in MatFormer [16]. We would like to point out that MatFormer’s mix’n’match happens on a per-layer basis, each layer processing all tokens uniformly using a particular nested model. Our method, in contrast, takes decisions per-token. MatFormer does not explicitly explore input-dependent dynamic decision-making for tokens. It is non-trivial to come up with a learning framework which would allow inference time mix’n’match where tokens are processed through different models.
>
> Regarding the concern about comparison with other MoE methods like Sparse VMoE, we discuss the comparison of MoNE with these MoE frameworks in the **MoE Comparison** section of the Global Author Rebuttal. In addition, MoNE can be extended to have multiple disjoint experts as in Sparse Vision-MoE, offering compute efficiency in such models. This can be considered as a future work. We will update the manuscript with this clarification.
>
> **Clarifying the Advantages of Nested Experts and Comparison with Dynamic Token Processing Approaches**
>
> The key advantage of having a nested set of experts over non-overlapping experts is that the computational gains can be achieved without increasing the parameter space. Non-overlapping experts for a given parameter space would lead to a limitation of representation power of each expert. However, as in MoNE, overlap between experts allows the largest expert to enjoy the full parameter space. Additionally, as shown in Table 5 of the MatFormer [16] paper, joint optimization of shared experts leads to better performance than having independent experts of the same size.
>
> We additionally compare our method with some of the methods mentioned by the reviewer, and show results in the **Baseline Comparisons** section of the Global Author Rebuttal. The comparisons show that MoNE performs better than other adaptive processing algorithms.
>
> **Understanding Expert Specialization: Contrasting MoNE and Conventional Sparse MoE Architectures**
>
> The primary motivation behind the nestedness in MoNE is input adaptivity thus offering compute efficiency while keeping the same parameter space. MoNE still remains fully compatible with traditional MoE approaches (Mixture of MoNE). In that setup, intuitively speaking, not all concepts may need the same amount of compute and using MoNE in the MoE setup would offer compute efficient MoE models.
>
> While one of the expected outcomes of MoE architectures is specialization of experts, this is not always the case in literature. Quoting a few lines from Section 5 of Mixtral of Experts [b], a MoE method that has non-overlapping experts - “Surprisingly, we do not observe obvious patterns in the assignment of experts based on the topic. For instance, at all layers, the distribution of expert assignment is very similar for ArXiv papers (written in Latex), for biology (PubMed Abstracts), and for Philosophy (PhilPapers) documents.“ Moreover, in Sparse VMoE [34] the authors observe very weak correlation of router decisions to categories.
>
> [b] A Jiang et al. Mixtral of Experts. arXiv:2401.04088, 2024.
>
> **Understanding the Impact of the Number of Routers in MoNE**
>
> This is a very important point raised by the reviewer and we clarify in the **Number of Routers** section of the Global Author Rebuttal.

---

> > ### Comment · Reviewer_E3Y8 · 2024-08-11
> >
> > Thank you for your detailed responses and the addition of new baseline comparisons. My main concerns have been addressed, and I am pleased to raise my score to 6.
> >
> > I would like to bring to your attention Flextron published at ICML'24 - "Flextron: Many-in-One Flexible Large Language Model" by Cai, Ruisi, et al. This work presents a very similar idea to your paper. While it is clearly a concurrent work and does not diminish the contributions of your paper, it would be beneficial to include a brief discussion highlighting the differences between the two approaches.

---

> > > ### Author Response · Authors · 2024-08-12
> > >
> > > Thank you for your valuable feedback and increased score. We are pleased to hear that your main concerns have been addressed.
> > >
> > > We appreciate you bringing Flextron to our attention. We have taken note of this concurrent work and agree that it would be beneficial to include a brief discussion in our paper highlighting the differences between the two approaches. We will make sure to incorporate this in our revised version.

---

### Author Rebuttal · Authors · 2024-08-07

**Latency/Throughput**

We present the latency/throughput gains of MoNE compared to baselines here, in addition to the FLOP gains mentioned in the paper. In the table below, we show absolute wall clock times and throughput for MoNE compared to a baseline ViViT model, on a single V100 GPU, achieving ~2x improvement in both FLOPs as well as runtime, whilst maintaining accuracy. We additionally show the variation of latency and throughput with FLOPs for varying model capacities $e_c$ of MoNE in Fig 1, 2 (attached pdf). The plots show that latency and throughput gains scales close to linearly with FLOP gains. Inference gains depend heavily on implementation. A simple high-level efficient implementation of our framework yields gains of this scale, and we believe that further improvements can be obtained by optimizing a low-level GPU kernel implementation for MoNE.

|Method|FLOPs (G)|Throughput (clips/sec)|Latency (ms)|Top-1 Acc|
|-|-|-|-|-|
| ViViT-FE-B/16| 376| 15.8| 129.2| 64.4|
| MoNE $e_c=0.3$|**162**| **30.7**| **65.5**|**64.6**|

**MoE Comparison**

Our framework MoNE, unlike traditional MoEs, does not increase the parameter space. Traditional MoEs, like Sparse VMoE, route inputs in each layer to one out of k independent experts (typically the FFN block), each having the same parameter footprint, thus increasing the parameter space k-fold for the expert blocks. On the other hand, independent MoNE blocks can potentially be used as experts in the MoE framework.  Therefore, our work _complements_ VMoE and other similar works. MoNE in this paper acts as an in-place replacement for a dense model (ViT), and hence all our comparisons maintain the same parameter space. VMoE frameworks show cross-scale results at the expense of increased parameter space (e.g., equivalent performance of VMoE-L/16 to ViT-H/14 in Table 2 in [34], and similar cross-scale comparisons in Figs. 4 to 8 in [1]). MoNE, in contrast, matches baseline performance with limited inference compute while working with the same parameter space.

**Baseline Comparisons**

We compare MoNE with more baselines, particularly with adaptive computation of dense models, as shown in the table below. We performed this experiment on ImageNet1k with a Ti/16 sized model. ACT, PonderNet, DepthAdapt, A-ViT are works with similar motivation of input adaptivity as MoNE, and MoNE shows superior performance. Latency gains on bigger models e.g., ViT-B in our paper) are even higher, as also observed in literature [7].

| Method|GFLOPs|Throughput (img/sec)|Top-1 Acc (ImageNet1k) |
|-|-|-|-|
| ViT|1.3|3410|71.3|
| PonderNet [a, 18]| 1.0| - | 66.2|
| Depth Adapt [18] | 1.1| - | 69.4|
| ACT [22]| 1.0| - | 71.0 |
| A-ViT [47]| 0.8| - | 71.0 |
| MoNE (Ours)| **0.8**| **4333**| **71.4**|

Additionally, in Fig. 3 (attached PDF), we apply Token Merging (ToMe) [7] on top of the MoNE style ViT-Ti/16 model trained on ImageNet1k. We trained a model with full capacity till layer 3 and a router with $e_c=0.5$ after that. We applied ToMe only on the first 3 layers. For fair comparison, we compare the performance drop and quote the same from a ViT-Ti model from the ToMe paper. Our preliminary results demonstrate that this implementation improves performance compared to ToMe on ViT, and this can be made better by extending this approach to all MoNE layers, applying it to distinct sets of nested tokens. This indicates that ToMe is complementary to MoNE. We will add these results to the revised manuscript.

[a] Banino et. al, PonderNet: Learning to Ponder, arXiv:2107.05407, 2021.

**Number of Routers**

Note that the number of routers does not have the same implications in MoNE as compared to traditional MoE. In MoEs, the parameter count increases with the number of layers on which the expert router is placed, and hence we typically see performance gains. Even then, in Sparse VMoE [34] (Table 2, 8), significantly increasing the parameter count with more routers only marginally improves performance. On the other hand, in MoNE, the parameter size remains fixed irrespective of the number of routers: the only change a router brings is re-assignment of tokens to nested experts while keeping the total compute per layer fixed. We hypothesize that increasing the number of routers leads to slight decrease in performance due to two reasons: (1) it brings in additional optimization challenges (as prevalent in the MoE literature as well [34]), (2) when we reassign a token from a smaller to bigger nested expert, its information content may be bounded by the representation power of the smaller expert, hence not improving performance. The opposite may happen from bigger to smaller nested experts, thus losing information. Since MoNE allows flexibility in the placement of routers, an interesting future direction would be to extend MoNE to more challenging task settings, where a higher number of routers might lead to better results. We will update the revision with this discussion.

**Task Difficulty based on router decisions**

We visualize the inputs that require the least and most compute per the router decisions, in a setting without capacity constraints in Fig. 4 (attached PDF). This is to understand if the router decisions correlate with task difficulty, i.e., send harder samples to larger experts. Specifically, instead of performing token to nested expert assignment using the EPR algorithm in the paper, we use the router decisions as is by just an argmax. Using this, we get an estimate of the computation that the router wants to assign to an image. The two sets of images indicate the top-3 images that demand the lowest and the highest compute, according to the router decisions. We observe that the images demanding less compute are visually simple, while the ones demanding highest compute are relatively complex.

---

> ### Author Response · Authors · 2024-08-12
>
> Dear Reviewers and Area Chairs
>
> We wholeheartedly thank you for your thoughtful reviews and the time you've dedicated to improving our work. We've carefully addressed all the points raised and submitted our rebuttal. We would greatly appreciate any further feedback or confirmation on our responses.

---

### Decision · Program_Chairs · 2024-09-25

**Decision:**

Accept (poster)

**Comment:**

The paper presents the "Mixture of Nested Experts" (MoNE) concept. It permits to dynamically route tokens in a priority order and thus enables to reduce inference-time compute compared to classic MoE methods. Reviewers appreciated the motivation and results. However, reviewers also raised concerns regarding the choice of baselines (MoE is missing), comparability of results (some methods are fine-tuned and others are trained from scratch) and other minor points. While the rebuttal was able to address many points, some concerns remained (additional baselines like MoE, proper and fair comparison to all methods). AC thinks those remaining concerns are important and strongly encourages the authors to revise the paper to point out/acknowledge those concerns such that future readers are aware. AC trusts that the authors will do so and recommends to accept.